# LEGACY: A LIGHTWEIGHT ADAPTIVE GRADIENT COMPRESSION STRATEGY FOR DISTRIBUTED DEEP LEARNING

## ABSTRACT

Distributed learning has demonstrated remarkable success in training deep neural networks (DNNs) on large datasets, but the communication bottleneck reduces its scalability. Various compression techniques are proposed to alleviate this limitation; often they rely on computationally intensive methods to determine optimal compression parameters during training and are popularly referred to as adaptive compressors. Instead of the hard-to-tune hyperparameters for adaptive compressors, in this paper, we investigate the impact of two fundamental factors in DNN training, the layer size of the DNNs and their training phases, to design a simple yet efficient adaptive scheduler for any compressors to guide the compression parameters selection. We present a **L**ightweight **E**fficient **G**r**A**dient **C**ompression strateg**Y** or LEGACY that, in theory, can work with any compression technique to produce its simple adaptive counterpart. We benchmark LEGACY on distributed and federated training, involving 6 different DNN architectures for various tasks performed on large and challenging datasets, including ImageNet and WikiText-103. On ImageNet training, by sending similar average data volume, LEGACY's adaptive compression strategies improve the Top-1 accuracy of ResNet-50 by $7\% - 11\%$, compared to the uniform Top-0.1% compression throughout the training. Similarly, on WikiText-103, by using our layer-based adaptive compression strategy and sending similar average data volume, the perplexity of the Transformer-XL improves $\sim 26\%$ more than the uniform Top-0.1% compression used throughout the training. We publish anonymized code at: https://github.com/LEGACY-compression/LEGACY.

## 1 INTRODUCTION

With the rise of digital data and extraordinary computing power, distributed learning on multiple computing nodes is vastly adapted to achieve optimal training performance for large deep neural networks (DNNs) You et al. (2018); Wongpanich et al. (2021); Xu et al. (2021a); Dutta et al. (2020). However, distributed training requires exchanging gradients between the nodes; the massive volume of the exchanged updates creates a communication bottleneck, and different compressed communication techniques (quantization Alistarh et al. (2017); Dettmers (2015); Bernstein et al. (2018), sparsification Dutta et al. (2020); Aji & Heafield (2017); Stich et al. (2018); Alistarh et al. (2018), low-rank Vogels et al. (2019), and hybrid Basu et al. (2019)) are designed to mitigate this problem.

Among these techniques, sparsifiers achieve baseline performance by only sending a small subset of the gradient components. Moreover, the over-parameterized nature of the DNN models creates sparse gradients during training Vaswani et al. (2019), e.g., NCF He et al. (2017) and DeepLight Deng et al. (2021) gradients consist of roughly 40% and 99% zero elements, respectively. Therefore, one can further sparsify these models in an efficient distributed training. The Top-$k$ Aji & Heafield (2017) sparsifier, which transmits only the $k$ largest elements, is widely utilized in distributed training. E.g., by communicating only 0.36% of the largest gradient elements of ResNet-50 He et al. (2016) trained on Imagenet Deng et al. (2009), Lin et al. (2018) achieves a baseline no compression performance. Nevertheless, almost a decade after being introduced by Aji & Heafield (2017) for gradient compression, there is no clear recipe for what $k$ to set for training different DNN models using the Top-$k$ sparsifier. While Top-$k$ sends fixed data volume in each training iteration,

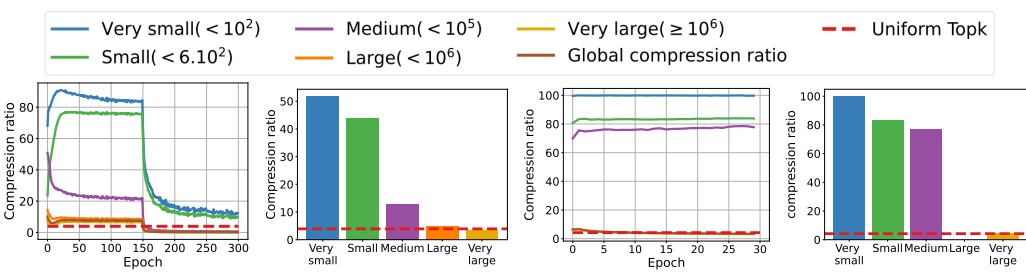

| (a) Compression by epoch | (b) Total compression | (c) Compression by epoch | (d) Total compression |

Figure 1: Compression ratio vs. the training iterations and layer size in training ResNet-18 on CIFAR-100 ((a) and (b)) and NCF on MovieLens-20 M ((c) and (d)) using the Top-$k$ and Threshold sparsifiers.

the threshold sparsifier (a.k.a. hard-threshold Strom (2015); Dutta et al. (2020); Sahu et al. (2021)) communicates gradient components with absolute magnitude greater than a threshold, $\lambda \geq 0$. It sets anything less than $\lambda$ to zero. This allows the threshold sparsifier to send a variable amount of data in each iteration and has a better convergence guarantee Sahu et al. (2021). One can see the threshold sparsifier as a *simple adaptive counterpart of Top-$k$* as it sends variable data volume in each training iteration. Although theoretically attractive, the same question persists — how to tune the threshold, $\lambda$ in practice?

Not only the sparsifiers, (or Top-$k$ in particular) regardless of the compressors, the existing literature predominantly focuses on uniform compression throughout the training, where the same compression ratio is used for all layers. Although varying the compression ratio for each layer at different stages of training is feasible, this area is not well-explored and most available literature proposes compute-heavy methods to find the best compressor Alimohammadi et al. (2023); Xin et al. (2023); Khirirat et al. (2021). Attempts were made to achieve optimal compression performance by adopting different adaptive strategies; see §2. In contrast, we investigated Occam's Razor principle: "plurality should not be posited without necessity." Instead of employing compute-intensive adaptive compressors, can we provide a simple, yet efficient strategy for quickly selecting a compression parameter for each layer, achieving a good balance between compressed data volume and model performance?

In that pursuit, we train two DNN architectures: (*i*) ResNet-18 He et al. (2016) on CIFAR 100 Krizhevsky et al. (2009) dataset (baseline no compression Top-1 accuracy is 73.38%) and (*ii*) NCF on MovieLens-20M dataset Harper & Konstan (2015) (baseline no compression best Hit-Rate@10 is 95.59%), on standard PyTorch benchmark using 2 NVIDIA A100-SXM4 GPUs with 80 GB memory, connected via 400 Gbps network bandwidth. We use the Top-$k$ and threshold sparsifiers and set the hyperparameters $k$ and $\lambda$ to send the same data volume. For ResNet18 and NCF, $k$ is set to 3.92%, and 0.35%, respectively, and $\lambda = 0.1$. While uniform Top-$k$ achieves a Top-1 accuracy of 73.04% on ResNet-18 and a best Hit-Rate@10 of 91.33% on NCF, threshold sparsifier achieves a Top-1 accuracy of 73.32% on ResNet-18 and a best Hit-Rate@10 of 92.7% on NCF, respectively. To get a better insight into threshold sparsifier's superior performance over the Top-$k$, in Figure 1 (a), we plot the compression ratio for different layers of ResNet-18 over iterations and in Figure 1 (b), we plot the total average compression of its different size layers. We observed that the small and medium layers (dimension less than $10^2$ to up to $10^5$) are not so severely compressed during the training compared to the large and very large layers (dimension more than $10^6$) — larger layers experience extremely aggressive compression — even more aggressive than the uniform Top-$k$ for those layers. Additionally, regardless of their sizes, during the beginning phase of the training, the layers are less aggressively compressed compared to the final training phase. We made almost identical observations in the NCF training; see Figures 1 (c)-(d).

Our empirical observations in using the Top-$k$ sparsifier and its adaptive counterpart for DNN training indicate *two key factors*: (*a*) *the layer size of the DNNs* influence in choosing how much one needs to compress, and (*b*) *the training phase of the DNNs* can be a critical contributor in the adaptive compressor design. Moreover, the second observation is consistent with recent research on the *critical training regime of DNNs* Achille et al. (2019); Zhang et al. (2022); Agarwal et al. (2021a). Although our quest for designing an adaptive compressor primarily started with sparsifiers, we believe, the above-mentioned simple factors can be used conjointly with any compression techniques in designing its compute-efficient, adaptive counterpart.

We list our contributions as follows:

**Adaptive compressor scheduler (§3).** We present a Lightweight Efficient GrAdient Compression StrategY or LEGACY that, in theory, can work with any compression technique to produce its simple adaptive counterpart. `LEGACY` is based on easy-to-obtain information — layer size and training phase. Designing `LEGACY` is empirically motivated and stands on solid technical intuitions; see §3.1. Irrespective of the DNN models and training dataset, `LEGACY` can guide the selection of compression parameters based on the layer size or training phase; see system design in §3.3.

**Theoretical insights (§4).** Under the usual assumptions for stochastic first-order algorithms in the compressed, distributed setup, we validate the influence of our policies on the convergence of compressed SGD; see Theorem 1 in §4.

**Benchmarking (§5).** We benchmark `LEGACY` through a variety of numerical experiments involving diverse DNN architectures (convolution and residual networks, transformer, and recommender system — a total of 6 models) trained for different tasks (image classification on CIFAR 10, CIFAR 100, and ImageNet, text prediction on WikiText-103, and collaborative filtering on Movielens-20M — a total of 5 datasets; see Table 2 in §B for a summary) by using Top-$k$ and Random-$k$ as base compressors. We report our results using multiple metrics: test accuracy, communicated data volume, throughput, and computation time. Additionally, we compared `LEGACY` against 4 state-of-the-art adaptive compressors (CAT Khirirat et al. (2021), Variance-based compression Tsuzuku et al. (2018), Accordion Agarwal et al. (2021a) and AdaComp Chen et al. (2018a)). Finally, in §5.5, we deploy `LEGACY` in a real federated training where the network bandwidth can pose a serious communication bottleneck.

## 2 RELATED WORK AND BACKGROUND

**Gradient compression techniques** are broadly divided into four classes: quantization Alistarh et al. (2017); Dettmers (2015); Bernstein et al. (2018); Wen et al. (2017), sparsification Aji & Heafield (2017); Stich et al. (2018); Alistarh et al. (2018), low-rank Vogels et al. (2019); Wang et al. (2018); Yu et al. (2018), and hybrid Strom (2015); Basu et al. (2019); Dryden et al. (2016).

**Adaptive compression in high-bandwidth data center.** The conventional practice employs the one-size-fits-all strategy, in which the compression parameters remain constant limiting the optimization potential and impacting model performance and communication resources. L-Greco Alimohammadi et al. (2023) utilizes dynamic programming to determine the optimal compression parameter for each layer under a fixed communication budget. Kimad Xin et al. (2023) and ACE Wang et al. (2024) dynamically monitors network bandwidth instead of using a fixed communication budget; CAT Khirirat et al. (2021) employs a communication cost model to optimize compression efficiency per communicated bit at each iteration. Inspired by the notion of a critical regime Achille et al. (2019), which emphasizes model sensitivity in a certain period, Accordion Agarwal et al. (2021a) aims to identify and respond to this regime by applying a lighter compression during the critical periods. Conversely, LAGS-SGD Shi et al. (2020), and COVAP Meng et al. (2023) take a different approach by adjusting the compression level to overlap gradient communications with computational tasks.

In less compute-intensive strategies, we list Luo et al. (2021) which decides the compression degree based on a probability that depends on the gradient value and the layer size. SDAGC Chen et al. (2020b) adjusts compression thresholds based on the standard deviation of gradients of each layer. AdaComp Chen et al. (2018a) is similar to the threshold compressor, divides gradient components into bins and selects significant components relative to the maximum value in each bin. Guo et al. (2020) determines the quantization level based on the gradient's mean-to-standard deviation ratio; DAGC Lu et al. (2023) assigns compression ratios to workers based on the data distribution. DLS Zhang et al. (2023a) tries to find a layer-wise Top-$k$ compression level. AdapTop-$k$ Ruan et al. (2023) sends more components at the beginning and end of the training and fewer components in the middle. Chen et al. (2018b); Wang et al. (2023; 2022); Deng et al. (2024) suggest freezing or skipping some layers based on their deviation from the previous iteration or by evaluating the importance of the learning of each layer. It can potentially reduce communication and computation by avoiding the gradient computation for first layers Miyauchi et al. (2018); Wang et al. (2022). Qu et al. (2024); Chen et al. (2020a) compress up and downlink communication.

**Algorithm 1:** Compressed distributed training without error feedback (EF)

**Input:** Number of nodes $n$, learning rate $\eta$, number of iterations $\mathtt{T}$, batch-size $\mathcal{B}$ per node as $\mathtt{n_{batch}}$
**Output:** The trained model $x$
**for** $t = 0, 1, \ldots, \mathtt{T}$ **do**
    **On each node** $i$:
    $g_{i,t} = \mathtt{Calculategradient}(x_t, \mathtt{n_{batch}})$
    $k_{i,t} = \mathtt{Chooseparam}(g_{i,t}, t)$
    $\tilde{g}_{i,t} = \mathtt{Compress}(g_{i,t}, k_{i,t})$
    $\mathtt{Communicate}(\tilde{g}_{i,t})$
    **On Master**:
    $[\tilde{g}_{1,t}, \ldots, \tilde{g}_{n,t}] = \mathtt{Receive}(\mathtt{n})$
    $[g_{1,t}, \ldots, g_{n,t}] = \mathtt{Decompress}([\tilde{\mathtt{g}}_{1,\mathtt{t}}, \ldots, \tilde{\mathtt{g}}_{\mathtt{n},\mathtt{t}}])$
    $g_t = \mathtt{AverageGrads}([g_{1,t}, \ldots, g_{n,t}])$
    $\mathtt{Broadcast}(\mathtt{g_t})$
    **On each node** $i$:
    $x_{t+1} = \mathtt{Update}(x_t, g_t, \eta)$

Table 1: Functions used in our framework.

| Function | Description |
|---|---|
| Chooseparam | Decide compression parameters |
| Compress | Apply compression to each layer |
| Communicate | Send compressed gradient to the server |
| Receive | Gather the compressed gradients from workers |
| Decompress | Restore the original tensor shape |
| AverageGrads | Average the received gradients |
| Broadcast | Broadcast the averaged gradient |
| Update | Optimizer independent parameter update |

**Transition to low-bandwidth network.** Compute-intensive techniques such as CAT Khirirat et al. (2021) face performance trade-offs, particularly in fast network environments like data centers Agarwal et al. (2021b). In such cases, using basic compressors without the extra burden of adaptive computations might take longer than no compression baselines Xu et al. (2021a); Eghlidi & Jaggi (2020); Zhang et al. (2023b). The scenario changes in federated learning (FL) Kairouz et al. (2019); Bergou et al. (2023); Xu et al. (2021b); Sun et al. (2024), where low-bandwidth heterogeneous network is de facto. Hence, compression becomes necessary; but employing complex adaptive compressors may reduce the data-saving advantages in FL, especially when weaker nodes are involved. As a result, we need to focus more on lightweight and simple principles to achieve adaptive compression.

**Notations.** We use $\|x\|$ to denote the $\ell_2$-norm of a vector $x$. By $g_{i,t}$ and $\nabla f_{i,t}$, we denote the stochastic gradient and full gradient, respectively, at the $i^{th}$ node at iteration $t$.

**Compressor.** A random operator, $\mathcal{C}(\cdot) : \mathbb{R}^d \to \mathbb{R}^d$ is a *compression operator* if $\mathbb{E}_{\mathcal{C}}\|x - \mathcal{C}(x)\|^2 \leq (1 - \delta)\|x\|^2$ for all $x \in \mathbb{R}^d$, where $\delta > 0$ is the compression factor. A smaller $\delta$ indicates a more aggressive compression. In our setup, $\delta \in (0, 1]$, and $\mathcal{C}$ is a $\delta$-*compressor*. The popular sparsifiers, $\mathrm{Top}_k$ and $\mathrm{Random}_k$ have $\delta = \frac{k}{d}$, and $\mathbb{E}\|x - \mathrm{Top}_k(x)\|^2 \leq \mathbb{E}\|x - \mathrm{Random}_k(x)\|^2 \leq (1 - \frac{k}{d})\|x\|^2$.

## 3 HOW CAN WE DESIGN AN ADAPTIVE COMPRESSOR SCHEDULER?

We observe two key factors in DNN training through the examples in Figure 1. First, the compression ratio has more impact at the beginning of the training than at the end. Second, regarding the topology of the considered network, it is better to compress large layers and keep small layers uncompressed (or with easy compression). But, can these observations also be theoretically justified so that we can build an adaptive compressor scheduler based on them?

To answer this, we formulate the impact of unbiased compressors on the decrease rate for the gradient descent (GD) algorithm under two relatively easier-to-analyze cases: (*i*) smooth, strongly convex functions, and (*ii*) smooth, nonconvex functions with PL condition. There is no loss of generality in considering GD instead of distributed SGD — analysis of GD offers ease of notations, and under simple arguments, leads us to a practical scheduler.

**Setup.** Consider the *empirical risk minimization* (ERM) problem with $n$ computing nodes:

$$\min_{x \in \mathbb{R}^d} \left[ F(x) := \frac{1}{n} \sum_{i=1}^{n} f_i(x) \right], \tag{1}$$

where $f_i(x) := \mathbb{E}_{z_i \sim \mathcal{D}_i} l(x; z_i)$ denotes the loss function evaluated at the $i^{th}$ node on input $z_i$ sampled from its distribution, $\mathcal{D}_i$. Let $g_{i,t}$ be the stochastic gradient computed at $i^{th}$ node at iteration $t$ and of the form $g_{i,t} = \nabla f_{i,t} + \xi_{i,t}$, with $\mathbb{E}[\xi_{i,t}|x_t] = 0$. To prove our results, we make some general assumptions; see §A.1.

**Function 1:** `EpochCompression`
$(\{\lambda_i\}_{i=1}^p, \{\delta_i\}_{i=1}^p)$

**Input:** Current iteration, $t$
**Output:** Compression parameter, $\delta_i$
$j =$ `index` of the smallest threshold from
  $\{\lambda_i\}_{i=1}^p$ such that iteration $t \leq \lambda_i$ ;
**return** $\delta_j$

**Function 2:** `LayerSizeCompression`$(\{\lambda_i\}_{i=1}^p, \{\delta_i\}_{i=1}^p)$

**Input:** Gradient $g_{i,t}$ at iteration $t$ from worker $i$
**Output:** `compression parameters list`
**for** *each layer L **in** $g_{i,t}$* **do**
  $j =$ `index` of the smallest threshold from
    $\{\lambda_i\}_{i=1}^p$ such that $|L| \leq \lambda_j$;
  `Append` $\delta_j$ to `compression parameters list`;
**return** `compression parameters list`;

## 3.1 Insight through the lens of the compressed GD algorithm

Let $\mathcal{C}_t$ be unbiased $\delta_t$-compressors for all $t \in [T]$. The iterative update rule of the compressed GD algorithm with fixed stepsize, $\eta \geq 0$ and unbiased $\delta_t$-compressors in solving (1) is given by

$$x_{t+1} = x_t - \eta \mathcal{C}_t(\nabla F(x_t)). \tag{2}$$

In the following lemma, we quantify the decrease in the quantity, $\|x_{t+1} - x_*\|^2$ under the smoothness and strong convexity assumption; see the proof in §A.2.

**Lemma 1.** *Let $F$ follow Assumptions 1 and 2. Then with fixed stepsize $\eta$, the sequence of iterates, $\{x_t\}_{t\geq 0}$ of compressed GD updates satisfy*

$$\mathbb{E}_{\mathcal{C}_t}\|x_{t+1} - x_*\|^2 \leq \underbrace{\left(1 - 2\mu\eta + \eta^2 \mu L(2 - \delta_t)\right)\|x_t - x_*\|^2}_{D(\delta_t):=Real\ decrease}.$$

Note that, the quantity $D(\delta_t)$ is a function of the compression factor. For no compression, $\delta_t = 1$, and we obtain:

$$\|x_{t+1} - x_*\|^2 \leq \underbrace{\left(1 - 2\mu\eta + \mu\eta^2 L\right)\|x_t - x_*\|^2}_{D(1):=Ideal\ decrease}.$$

Ideally, we are interested in $\delta_t \in (0, 1]$ such that $D(\delta_t)$ (i.e., the compressed GD decrease) is as close as possible to $D(1)$ (i.e., the non-compressed GD decrease). We have

$$\Delta := D(\delta_t) - D(1) = \mu\eta^2 L(1 - \delta_t)\|x_t - x_*\|^2.$$

Therefore, to have $\Delta \approx 0$, we require: (*i*) At the beginning of the training, we have $\|x_t - x_*\|^2 \gg 0$. Therefore, to make $\Delta \approx 0$ we need to choose $\delta_t$ close to 1 (no or easy compression). (*ii*)At the end of the training we have $\|x_t - x_*\|^2 \approx 0$. Therefore, no strong control is needed on $\delta_t$ to keep $\Delta$ small. In this case, one can choose $\delta_t \approx 0$ (aggressive compression). Moreover, large layers contribute more significantly to $\|x_t - x_*\|^2$ compared to the small layers. Therefore, to keep $\Delta$ small, it is necessary to compress large layers more aggressively than the smaller ones.

To further extend our theoretical insight, in the next lemma, we consider GD for minimizing smooth nonconvex function under the PL condition and quantify the functional suboptimality gap, $E_{\mathcal{C}_t}(F_{t+1}) - F_*$; see the proof in §A.2.

**Lemma 2.** *Let $F$ follow Assumptions 1 and 4. Then with stepsize $\eta = \frac{1}{L}$, the sequence of iterates, $\{x_t\}_{t\geq 0}$ of compressed GD updates satisfy*

$$E_{\mathcal{C}_t}(F_{t+1}) - F_* \leq \underbrace{\left(1 - \frac{\delta_t \mu}{L}\right)(F_t - F_*)}_{D(\delta_t):=Real\ decrease}.$$

As before, substituting $\delta_t = 1$ gives the ideal decrease i.e., the decrease in the functional suboptimality gap without compression:

$$F_{t+1} - F_* \leq \underbrace{\left(1 - \frac{\mu}{L}\right)(F_t - F_*)}_{D(1):=Ideal\ decrease}.$$

To have $D(\delta_t) - D(1) = (1 - \delta_t)\frac{\mu}{L}(F_t - F_*) \approx 0$, we require: (*i*) At the beginning of the training $F_t - F_* \gg 0$. Therefore, we need to choose $\delta_t \approx 1$ (no or easy compression) to keep $D(\delta_t) - D(1) \approx 0$. (*ii*) At the end of the training $F_t - F_* \approx 0$. Therefore, we can choose $\delta_t \approx 0$ (aggressive compression).

Figure 2: System architecture. The `LEGACY` framework is highlighted in blue.

### 3.2 AN ADAPTIVE COMPRESSOR SCHEDULER FOR DNN TRAINING

Motivated by the previous section, we formally define an adaptive compressor scheduler for compressed distributed training on $n$ workers. Although our scheduler is optimizer agnostic, for simplicity, we consider the optimizer to be SGD. Given a stepsize sequence, $\{\eta_t \geq 0\}_{t \geq 0}$ and $\delta_t$-compressors, the update rule for compressed distributed SGD on $n$ workers is given by

$$x_{t+1} = x_t - \frac{\eta_t}{n} \sum_{i=1}^{n} \mathcal{C}_t(g_{i,t}). \tag{3}$$

Algorithm 1 provides a general compressed communication framework without error feedback Karimireddy et al. (2019). We build our approaches around the general framework of Algorithm 1, by changing the compression level through function `chooseparam`. We require two user-inferred hyperparameters: (*i*) a sorted list of $p$ decreasing compression levels, $\{\delta_i\}_{i=1}^{p}$, of the $\delta$-compressor $\mathcal{C}_t$, where $\delta_p$ being the most aggressive compression factor, and (*ii*) a sorted list of $p$ non-decreasing thresholds, $\{\lambda_i \geq 0\}_{i=1}^{p}$, which represents either an iteration or a layer size at which we use a certain compression level $\delta_i$, in Algorithm 1. The threshold change is based on the following approaches:

(*i*) **Training epoch dependent.** We start with a less intense compression and gradually increase its intensity during the training. In `Epoch compression`, we progressively increase the compression level $\delta$ as training progresses; see Function 1. In this case, the non-decreasing thresholds $\{\lambda_i\}_{i=1}^{p}$ denote the iterations or epochs at which the compression intensity is increased.

(*ii*) **Layer size dependent.** We employ an easy compression level for small layers as their size is insignificant compared to the larger ones. We achieve this through `Layer size compression`; see Function 2. In this Function, we used the thresholds $\{\lambda_i\}_{i=1}^{p}$ to group layers by their sizes — smaller layers are affected by a less intense compression, while the larger layers experience a more aggressive compression.

### 3.3 SYSTEM ARCHITECTURE — LEGACY

We present Lightweight Efficient GrAdient Compression StrategY or `LEGACY`; see the system architecture in Figure 2. `LEGACY` is compatible with any machine learning framework (e.g., TensorFlow, PyTorch), and offers a simple API that can be embedded with various gradient compressors (e.g., Top-$k$, QSGD, etc.). In this work, we use sparsifiers as *base compressors* in `LEGACY` and use NCCL `AllGather` communication collective NCCL. However, `LEGACY` is agnostic of the optimizer used for training and it can be effortlessly integrated with other communication protocols such as P2P or `AllReduce` communication collective.

For transmitting workers, `LEGACY` is executed through the intermediary API call `chooseparam` in Algorithm 1, responsible for selecting the appropriate compression parameters for each layer. After gradient computation through any ML benchmark, based on the user's strategy, epoch compression Function 1 (⚑ = 1) or Layer size compression Function 2 (⚑ = 0) is invoked to dynamically determine the compression parameters for each layer, which are then applied to the gradient compressor in the worker. Additionally, Functions 1 and 2 in `LEGACY` can be used conjointly with the base-compressor; see the blue three-point arrow. Other than `chooseparam`, `LEGACY` uses other well-known APIs for communication, averaging, broadcasting, etc. from the GRACE library Xu et al. (2021a); see Table 1. The receiving worker does not require any modulation, it applies reverse operations and decompresses the received gradient. In master-worker formalization, `LEGACY` can be used for uplink and downlink bidirectional compression.

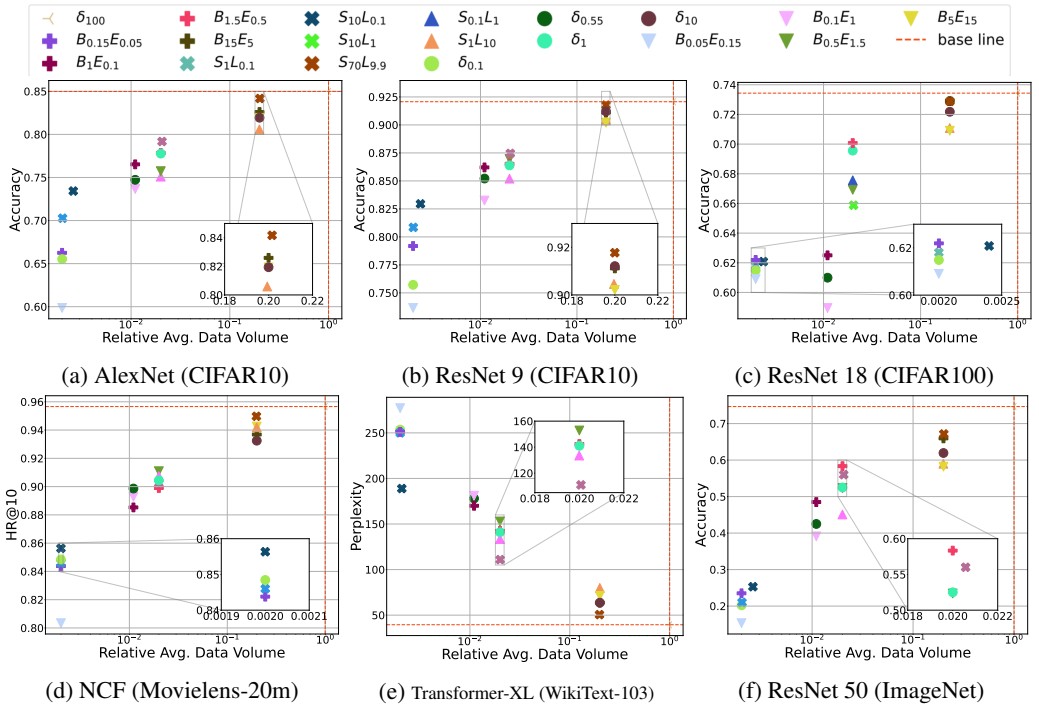

Figure 3: Layer-size and training epoch dependent Top-$k$ and uniform Top-$k$ (denoted by only $\delta_{\text{compression}}$) — Relative average data volume vs. model quality.

## 4 CONVERGENCE GUARANTEE

Inspired by Sahu et al. (2021); Stich & Karimireddy (2020), we establish the nonconvex convergence of distributed SGD with $\delta_t$-compression operators, $\mathcal{C}_t$. Ideally, we want the compressed stochastic gradient steps to be as close as possible to the full gradient and the descent on the optimization objective to be as close as possible to that of the one without compression. This implies we are interested in *minimizing* $\mathbb{E}\left[\left\|\frac{1}{n}\left(\sum_{i=1}^{n}\mathcal{C}_t(g_{i,t}) - \sum_{i=1}^{n}\nabla f_{i,t}\right)\right\|^2 |x_t\right]$. We measure this deviation in the following result. Denote $\beta_t := (1-\delta_t)(M+1) + M$, where $M, \sigma^2 \geq 0$ are constants such that for all $x_t \in \mathbb{R}^d$, the stochastic noise, $\xi_{i,t}$ follows $\mathbb{E}[\|\xi_{i,t}\|^2 \mid x_t] \leq M\|\nabla f_{i,t}\|^2 + \sigma^2$; see Assumption 5. The constants appearing in our results are due to the general Assumptions in §A.1.

**Lemma 3.** *(Compression variance) Let $\mathcal{C}_t$ be $\delta_t$-compressors for all $t \in [T]$, and let $F$ follow Assumption 6, and the stochastic noise follow Assumption 5. We have*

$$\mathbb{E}\left[\left\|\tfrac{1}{n}\left(\sum_{i=1}^{n}\mathcal{C}_t(g_{i,t}) - \sum_{i=1}^{n}\nabla f_{i,t}\right)\right\|^2 |x_t\right] \leq \tfrac{\beta_t}{n}\left(2A(F_t - F_\star) + B + \|\nabla F_t\|^2\right) + \tfrac{(2-\delta_t)\sigma^2}{n}.$$

Using the previous Lemma, the following theorem gives the complexity results, which are similar to the classical complexity results for compressed SGD type of algorithms; see Dutta et al. (2020); Stich & Karimireddy (2020); Sahu et al. (2021). See the detailed proof in §A.3.

**Theorem 1.** *(Nonconvex convergence) Let Assumptions 1, 5, and 6 hold. Let $\mathcal{C}_t$ be $\delta_t$-compressors for all $t \in [T]$. For a stepsize $\eta \leq \min\left(\frac{1}{\frac{L}{2}+\frac{L(2M+1)}{n}}, \left(\frac{AL(2M+1)T}{n}\right)^{-\frac{1}{2}}\right)$ we have:*

$$\min_{t=0,1,\cdots T-1}\mathbb{E}\|\nabla F_t\|^2 \leq \tfrac{3}{T\eta\left(1-\frac{L\eta}{2}-\frac{L\eta}{n}\right)}\left(F_0 - F_\star\right) + \tfrac{L\eta\left(B(2M+1)+2\sigma^2\right)}{2n\left(1-\frac{L\eta}{2}-\frac{L\eta(2M+1)}{n}\right)}.$$

## 5 BENCHMARKING AND EVALUTAION

**Environment and Configuration.** We run our experiments on 4 NVIDIA A100-SXM4 GPUs (2 GPUs for AlexNet, ResNet-9, and ResNet-18 training, and 4 GPUs for Transformer-XL, NCF, and

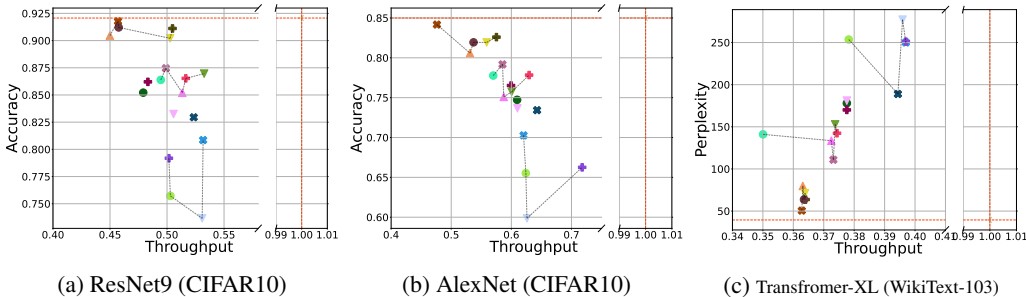

(a) ResNet9 (CIFAR10)     (b) AlexNet (CIFAR10)     (c) Transfromer-XL (WikiText-103)

Figure 4: Layer-size and training epoch dependent Top-$k$ and uniform Top-$k$ (denoted by only $\delta$) — Throughput vs. model quality, where experiments with similar global compression ratios are linked with a dotted line; see Legend in Figure 3.

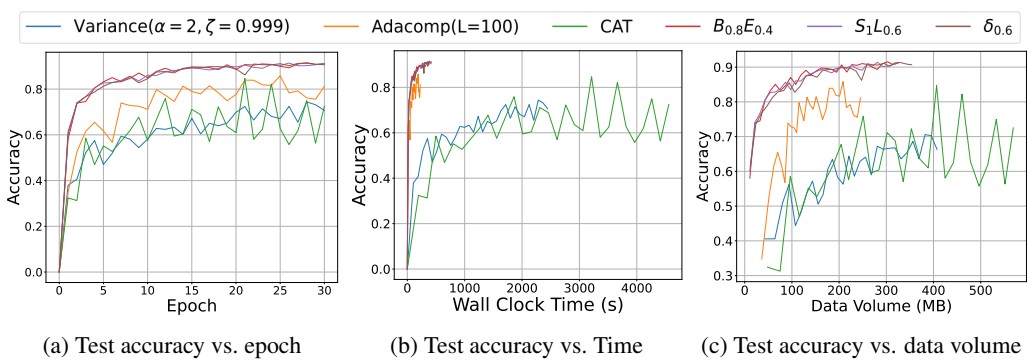

(a) Test accuracy vs. epoch     (b) Test accuracy vs. Time     (c) Test accuracy vs. data volume

Figure 5: Comparison with the state-of-the-art adaptive compressors in training ResNet9 on CIFAR10.

ResNet-50 training) with 80GB memory and interconnected with 400 GBps bandwidth. LEGACY is built on Dutta et al. (2020); Sahu et al. (2021); for Transformer-XL, we utilized the NVIDIA Training Examples benchmark Nvidia with reduced steps; tests on CIFAR10, CIFAR100, and NCF were implemented using Dutta et al. (2020), Sahu et al. (2021), and Nvidia, respectively. We used 30 epochs for AlexNet, ResNet-9, and NCF training, 300 epochs for ResNet18 training, and 4,500 steps for the Transformer training. For ImageNet, we employed PyTorch and train ResNet-50 for 50 epochs; see Tables 2 and 3 in §B for a detailed summary. For experimental reproducibility see §B.5.

**LEGACY Setup.** We split the training into two phases: beginning $B$ (first half of the total epochs) and end $E$ (rest of the total epochs); each phase uses a different compression level. For layer sizes, we categorize layers into two groups: small layers, $S$ with fewer than $10^4$ elements, and large layers, $L$ with $10^4$ elements or more. With this formalization, $S_{\delta_1} L_{\delta_2}$ means small layers are compressed with compression factor, $\delta_1$ and large layers compressed with compression factor, $\delta_2$, and $B_{\delta_1} E_{\delta_2}$ denotes two-phase training, beginning phase with compression factor, $\delta_1$, and end phase with $\delta_2$.

## 5.1 MODEL QUALITY VS. TRANSMITTED DATA VOLUME

Figure 3a shows the accuracy of AlexNet on CIFAR-10; uniform Top-$k$ compression with $k = 0.1\%d$ (corresponding to the $\delta_{0.1}$) results in an accuracy of 75.7%. However, using Top-$k$ as base compression in LEGACY, the strategy, $B_{0.15} E_{0.05}$, which starts with a compression ratio of 0.15% for the first half of the epochs and then switches to an aggressive compression ratio of 0.05%, achieves a higher accuracy of 79.18%. Notably, the reverse strategy $B_{0.05} E_{0.15}$ results in a lower accuracy of 73.6%. When we compress smaller layers at 1% while keeping the larger layers at the 0.1% ratio, $S_1 L_{0.1}$, the accuracy improves by 5.14% over the uniform compression. Figures 3b – 3f show similar results across different DNN models and challenging, larger datasets including ImageNet and WikiText, with accuracy improvements up to 7-11% on ImageNet compared to the uniform compression strategy. For language model in Figure 3e, the perplexity improves $\sim 26\%$, from 253.57 with uniform $\delta_{0.1}$ to 188.8 with adaptive compression $S_{10} L_{0.1}$.

**Takeaways.** In general, sending more data leads to a better-trained model. However, for (almost) equally transmitted data volume, the results reveal that beginning the training with no to mild compression and transitioning to a more aggressive compression, yields better performance than using

a uniform or inverse compression strategy. DNN models retain more crucial information during the initial training phases starting with a mild compression and gradually increasing the compression ratio. This strategy allows the model to learn effectively from the data, leading to improved accuracy compared to a uniform compression strategy, or when aggressive compression is applied first and eased off. Gradually increasing the compression factor balances the need for sufficient data in the early stage and gains efficiency from higher compression in the later stage. Similar conclusions hold for layer size-dependent adaptive compression. Leaving small layers uncompressed or lightly compressed results in a minor increase in transmitted data volume while improving perplexity by 26% on WikiText-103 and accuracy by 7% on ImageNet.

## 5.2 MODEL QUALITY VS. TRAINING THROUGHPUT

Figure 4 shows the impact of compression on model quality as a function of the relative throughput. Test cases with a similar average compression ratio ($\pm 10\%$) are connected with dotted lines. The throughput from compression is inferior to the no-compression baseline as we use a limited number of workers in a data center connected by fast network bandwidth, and the overhead of compression could be higher relative to the network throughput. Analyzing the groups (connected by the dotted lines), we observe that the average compression ratio influences the model performance and throughput; sending more data results in higher model quality but lower throughput. For a similar average compression ratio, applying moderate compression during the initial training phase and to smaller layers yields better performance. In Figure 4a, for ResNet9, a uniform Top-0.1% compression results in 75% accuracy, and 50.29% relative throughput. However, our epoch-based strategy, $B_{0.15}E_{0.05}$ in LEGACY, yields similar relative throughput but improved accuracy, reaching 79.18%. Meanwhile, the layer size-based adaptive strategy, $S_1L_{0.1}$ in LEGACY, achieves better throughput at 53.16% and higher accuracy of 80.85% achieving a 5.7% gain in throughput and 6.6% gain in accuracy compared to the uniform compression. We observe similar findings in Figures 4b and 4c. Generally, the adaptive strategies in LEGACY (denoted by '+' for epoch-based and '×' for layer size-based) for linked points are positioned either above (indicating better accuracy) or to the right (indicating better throughput) of the uniform case for AlexNet and ResNet9. For the Transformer-XL, LEGACY strategy points are located to the right of or below the uniform case, under similar average compression ratios, indicating a better perplexity, with improvements of up to ~26% in perplexity and ~4.5% in throughput compared to uniform compression.

**Takeaways.** Our layer-based strategy can increase accuracy and throughput compared to the uniform or inverse approaches, although the throughput gains are limited due to the high-speed network in the data center. For the layer size-based approach, not compressing small layers eliminates the computational overhead. For the epoch-based approach, sending more data at the beginning appears to balance out the aggressive communication towards the end, yielding similar throughput while leveraging the early training stages to achieve better accuracy.

## 5.3 ADDITIONAL BENCHMARKING AND DISCUSSIONS

We use Random-$k$ as the base compressor in LEGACY and provide accuracy vs. data volume results; see in §B, Figure 7. In §B Table 4, we report the average Top-1 test accuracy of ResNet9 and AlexNet on CIFAR10, derived from 15 independent runs; the results are in agreement with §5.1. By using Top-$k$ as the base compressor (with and without error feedback) in LEGACY, we provide the model quality vs. wall clock time results in §B.3. See the limitations and social impact in §C.

## 5.4 COMPARISON WITH ADAPTIVE GRADIENT COMPRESSORS

We evaluate our approaches, $B_{\delta_1}E_{\delta_2}$ and $S_{\delta_1}L_{\delta_2}$, using the Top-$k$ compression in LEGACY against three state-of-the-art adaptive compression methods (Adacomp Chen et al. (2018a), variance-based compression Tsuzuku et al. (2018), and CAT Khirirat et al. (2021)) in terms of the trained model quality and the training time. From Figures 5a and 5b we observe the superior performance of our scheduler in terms of accuracy at similar exchanged data volumes. Although we experience slower training compared to Adacomp, as Adacomp is based on thresholding and $2\times$ faster than the uniform Top-$k$ and our strategies, we have a 12% accuracy gain than Adacomp by sending slightly over 75Mb more data; see Figure 5c. Variance-based compression requires access to per-sample gradients, which are not supported by most deep learning frameworks; obtaining these values using a batch size of one is extremely slow. We used OPACUS Yousefpour et al. (2021) to get faster per-sample gradients. Still, it remains $\sim 6\times$ slower than our approaches with a 15% lower

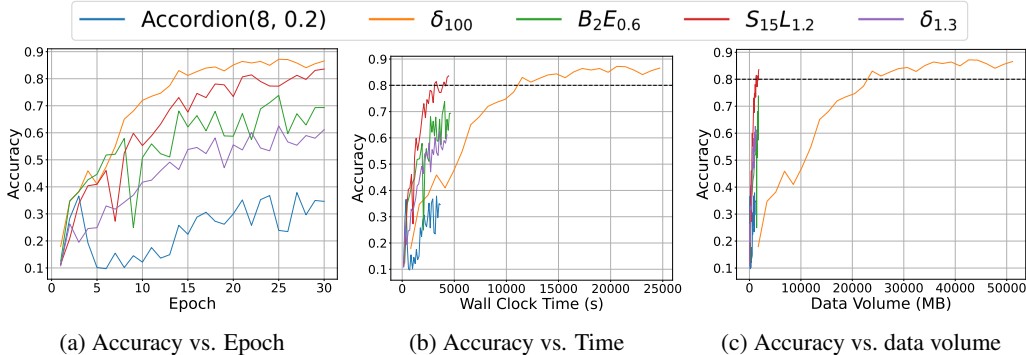

(a) Accuracy vs. Epoch     (b) Accuracy vs. Time     (c) Accuracy vs. data volume

Figure 6: Training ResNet18 on CIFAR10 in a FL Environment; $\delta_{100}$ is no compression baseline.

accuracy. CAT requires testing many values at each iteration before choosing the sparsity, resulting in $11\times$ slower performance, sending around 575Mb data, and incurring 25% lower accuracy than our approaches. Our strategies are robust as they choose the compression ratios and control the total and per-iteration data volume. In contrast, except Accordion, other adaptive methods can neither be applied to different compressors nor provide an estimate of the data volume. We also found that at the core, these methods exhibit similar behavior to our strategies, confirming the effectiveness of our approach, which does not require additional computation.

## 5.5 FEDERATED TRAINING OF RESNET-18 ON CIFAR-10

High network bandwidth generally does not harvest the benefit of compression Xu et al. (2021a); bandwidth-limited federated training is an authentic area in assessing our strategies.

**Testbed and setup.** We emulate an environment of 50 workers connected via 1Gbps network operating on Intel Xeon Platinum 8276 CPUs, instead of GPUs. Additionally, we partitioned the CIFAR10 dataset into 50 subsets using a Dirichlet distribution with parameter $\alpha = 10$, to mimic a non-i.i.d. data distribution among the workers. We use Top-$k$ as the base compressor in LEGACY and compare the results with no compression baseline and Accordion Agarwal et al. (2021a). We use Gloo `AllGather` for internodal communication. This configuration more accurately reflects the limitations encountered in a real-world FL environment, characterized by heterogeneous data, constrained networks, and computational resources.

**Result.** We do not accumulate gradients at local nodes but communicate immediately to test the resilience of training when the slow network is burdened with heavy communication. Our strategies are robust in FL and outperformed the uniform Top-$1.3\%$ and Accordion compressors, achieving a 16%-35% gain in accuracy, while being $6\times$ faster than the no-compression baseline; see Figure 6b. The test accuracy of our layer-based policy is almost similar to the no-compression baseline, while the epoch-based policy outperforms the uniform Top-$1.3\%$ compression. The adaptive policies in LEGACY significantly lower the communicate data volume overhead in FL deployments; $B_2E_{0.6}$ and $S_{15}L_{1.2}$ communicate only 1.3% and 1.23% of the data, respectively, compared to the no-compression baseline (Figure 6c); also, see total communicated data volume during training in Figure 8c. Together, this indicates the high quality of the trained model, consistent with the findings in data center training, and validates our claim that the simple yet efficient principles in LEGACY are beneficial for federated deployments.

## 6 CONCLUSION

This paper introduces a lightweight, adaptive gradient compression framework or LEGACY for distributed deep neural network training. LEGACY is open-source and can be seamlessly integrated into any ML framework. In contrast to the compute-intensive, parameter-heavy adaptive compressors, LEGACY operates based on two fundamental factors in DNN training, the layer size of the DNNs and their training phases, and provides a simple yet efficient adaptive scheduler for any compressors to guide their compression parameters selection. Our benchmarking of LEGACY using Top-$k$ and Random-$k$ as base compressors shows consistent performance gains compared to the uniform Top-$k$, Random-$k$, and four other state-of-the-art adaptive compressors across large and challenging datasets, including ImageNet and WikiText-103. Finally, in the bandwidth-constrained federated training, we profile the efficacy of LEGACY and establish the need of a simple, adaptive scheduler.

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

CONTENTS

A  THEORETICAL RESULTS

This section complements Sections 3 and 4 in the main paper. We start with the Assumptions used in the main paper.

## A.1 Assumptions

We make the following general assumptions.

**Assumption 1.** *(Smoothness) The loss function $f_i : \mathbb{R}^d \to \mathbb{R}$ at each node $i \in [n]$ is $L$-smooth, i.e. $f_i(y) \leq f_i(x) + \langle \nabla f_i(x), y - x \rangle + \frac{L}{2}\|y - x\|^2$ for all $x, y \in \mathbb{R}^d$.*

**Assumption 2.** *($\mu$-strongly convex) The loss function $f_i : \mathbb{R}^d \to \mathbb{R}$ at each node $i \in [n]$ is $\mu$-strongly convex, i.e. $f_i(y) \geq f_i(x) + \langle \nabla f_i(x), y - x \rangle + \frac{\mu}{2}\|y - x\|^2$ for all $x, y \in \mathbb{R}^d$.*

**Remark 1.** *The above two assumptions together imply that $F$ is $L$-smooth and $\mu$-strongly convex.*

**Assumption 3.** *(Global minimum) There exists $x_\star$ such that, $F(x_\star) = F_\star \leq F(x)$, for all $x \in \mathbb{R}^d$.*

**Assumption 4.** *(Polyak-Lojasiewicz Condition) The function $F$ satisfies Polyak-Lojasiewicz (PL) condition with parameter $\mu \geq 0$ if for all $x \in \mathbb{R}^d$ the following holds:*

$$\frac{1}{2}\|\nabla F(x)\| \geq \mu(F(x) - F_*).$$

**Assumption 5.** *($(M, \sigma^2)$ bounded noise) There exist constants $M, \sigma^2 \geq 0$, such that for all $x_t \in \mathbb{R}^d$, the stochastic noise, $\xi_{i,t}$ follows*

$$\mathbb{E}[\|\xi_{i,t}\|^2 \mid x_t] \leq M\|\nabla f_{i,t}\|^2 + \sigma^2.$$

**Remark 2.** *The above implies, $\mathbb{E}[\|g_{i,t}\|^2 \mid x_t] \leq (M + 1)\|\nabla f_{i,t}\|^2 + \sigma^2$.*

**Assumption 6.** *(Bounded variance of gradients) There exist constants $A, B \geq 0$ such that, for all $x \in \mathbb{R}^d$, the variance of gradients among nodes follow*

$$\frac{1}{n}\sum_{i \in [n]}\|\nabla f_i(x) - \nabla F(x)\|^2 \leq 2A(F(x) - F_\star) + B.$$

We impose the following extra assumption on the expected direction of the compressed gradient for biased compressors. A similar assumption was made in Dutta et al. (2020) and it has been validated by several classic biased compressors, such as Topk-$k$.

Let $\mathcal{C}$ be a biased $\delta$-compressor such that there exists $0 < \alpha \leq 2$ and $\beta > 0$ such that if $g \in \mathbb{R}^d$, is an unbiased estimator of $\nabla f$ then

$$\mathbb{E}\left[\mathcal{C}(g)^\top \nabla f | \nabla f\right] \geq \beta \mathbb{E}\|\nabla f\|^\alpha + R,$$

where $R$ is a small scalar residual which may appear due to the numerical inexactness of some operators or due to other computational overheads.

**Remark 3.** *The above assumption is general and one can characterize many compressors with this. For instance, for Top-k, we have $\alpha = 2$, $\beta = k/d$ and $R = 0$. In this paper, for simplicity and without loss of generality, we consider $\alpha = 2$, $\beta = 1$, and $R = 0$. Under these simplifications, the previous assumption aligns with the unbiasedness assumption of the compressor and the stochastic gradient $g$. Therefore, the convergence analysis is based on this assumption.*

### A.1.1 Inequalities used

1. If $a, b \in \mathbb{R}^d$ then we use a relaxed version of Peter-Paul inequality:

$$\|a + b\|^2 \leq 2\|a\|^2 + 2\|b\|^2. \tag{4}$$

2. If $a, b \in \mathbb{R}^d$ then the following holds:

$$2\langle a, b \rangle \leq 2\|a\|^2 + \frac{1}{2}\|b\|^2. \tag{5}$$

3. For $x_1, \ldots, x_n \in \mathbb{R}^d$ we have:

$$\|\sum_{i=1}^n x_i\|^2 \leq n \sum_{i=1}^n \|x_i\|^2. \tag{6}$$

4. If $X$ is a random variable then:

$$\mathbb{E}\|X\|^2 = \|\mathbb{E}[X]\|^2 + \mathbb{E}[\|X - E[X]\|^2]. \tag{7}$$

**Lemma 4.** *Let $\mathcal{C}(\cdot) : \mathbb{R}^d \to \mathbb{R}^d$ be a $\delta$-compressor. We have $\mathbb{E}\|\mathcal{C}(g)\|^2 \leq (2 - \delta)\|g\|^2$.*

*Proof.* Recall for $\delta$-compressors, we have $\mathbb{E}\|g - \mathcal{C}(g)\|^2 \leq (1 - \delta)\|g\|^2$. Since $\mathbb{E}(\mathcal{C}(g)) = g$, from equation 7 we have,

$$\mathbb{E}\|\mathcal{C}(g)\|^2 \overset{\text{By } equation\ 7}{=} \mathbb{E}\|g - \mathcal{C}(g)\|^2 + \|g\|^2 \leq (1 - \delta)\|g\|^2 + \|g\|^2 = (2 - \delta)\|g\|^2.$$

$\square$

**Lemma 5.** *Let $F$ follow Assumption 6. Then we have for all $t \geq 0$,*

$$\frac{1}{n} \sum_{i=1}^{n} \|\nabla f_{i,t}\|^2 \leq 2A(F_t - F_\star) + B + \|\nabla F_t\|^2. \tag{8}$$

*Proof.* The proof follows from the fact that $\frac{1}{n}\sum_{i=1}^{n}\|\nabla f_{i,t}\|^2 = \frac{1}{n}\sum_{i=1}^{n}\|\nabla f_{i,t} - \nabla F_t + \nabla F_t\|^2$ and $F_t := \frac{1}{n}\sum_{i=1}^{n} f_{i,t}$ for all $t \geq 0$. Therefore,

$$\frac{1}{n}\sum_{i=1}^{n}\|\nabla f_{i,t}\|^2 \quad = \quad \frac{1}{n}\sum_{i=1}^{n}\|\nabla f_{i,t} - \nabla F_t + \nabla F_t\|^2$$

$$= \quad \frac{1}{n}\sum_{i=1}^{n}\|\nabla f_{i,t} - \nabla F_t\|^2 + \|\nabla F_t\|^2$$

$$\overset{\text{By Assumption 6}}{\leq} \quad 2A(F_t - F_\star) + B + \|\nabla F_t\|^2.$$

Hence the result. $\square$

## A.2 Convergence of GD

This section provides the convergence proofs GD on strongly convex and nonconvex functions with PL conditions as given in Lemma 1 and Lemma 2.

### A.2.1 Convergence of GD on strongly convex functions

**Lemma 1.** *(**Gradient descent with unbiased compressor**) Let $F$ follow Assumptions 1 and 2. Then with stepsize $\eta \leq \frac{1}{(2-\delta_t)L}$, the sequence of iterates, $\{x_t\}_{t\geq 0}$ of compressed GD updates satisfy*

$$E_{C_t}(\|x_{t+1} - x_\star\|^2) \leq \left(1 - 2\mu\eta + \eta^2\mu L(2 - \delta_t)\right)\|x_t - x_\star\|^2. \tag{9}$$

*Proof.* From the GD update in equation 2, we have

$$x_{t+1} - x_\star = x_t - x_\star - \eta\mathcal{C}_t(\nabla F(x_t)).$$

Squaring both sides and expanding we have

$$\|x_{t+1} - x_\star\|^2 = \|x_t - x_\star\|^2 - 2\eta\mathcal{C}_t\left(\nabla F_t\right)^T\left(x_t - x_\star\right) + \eta^2\|\mathcal{C}_t(\nabla F_t)\|^2.$$

By taking expectation on the randomness of the compressors $\mathcal{C}_t$ we get:

$$\mathbb{E}_{\mathcal{C}_t}\left(\|x_{t+1} - x_\star\|^2\right) \quad = \quad \|x_t - x_\star\|^2 - 2\eta\nabla F_t^T(x_t - x_\star) + \eta^2\mathbb{E}_{\mathcal{C}_t}\|C_t(\nabla F_t)\|^2$$

$$\overset{\text{By Assumption 2}}{\leq} \quad \|x_t - x_\star\|^2 + 2\eta\left(F_\star - F_t\right) - \mu\eta\|x_t - x_\star\|^2$$
$$+ \eta^2(2 - \delta_t)\|\nabla F_t\|^2$$

$$\overset{\text{By Assumption 1}}{\leq} \quad \|x_t - x_\star\|^2 + 2\eta\left(F_\star - F_t\right) - \mu\eta\|x_t - x_\star\|^2$$
$$+ 2\eta^2 L(2 - \delta_t)(F_t - F_\star)$$

$$\leq \quad (1 - \mu\eta)\|x_t - x_\star\|^2 + 2\eta\left(\eta L(2 - \delta_t) - 1\right)(F_t - F_\star)$$

$$\overset{\text{By Assumption 2}}{\leq} \quad (1 - \mu\eta)\|x_t - x_*\|^2 + \mu\eta\left(\eta L(2 - \delta_t) - 1\right)\|x_t - x_*\|^2$$

$$\leq \quad \left(1 - 2\mu\eta + \eta^2\mu L(2 - \delta_t)\right)\|x_t - x_\star\|^2.$$

This completes the proof. $\square$

### A.2.2 Convergence of GD on nonconvex functions with PL condition

**Lemma 2.** *(Gradient descent with unbiased compressor) Let $F$ follow Assumptions 1 and 4. Then with stepsize $\eta = \frac{1}{L}$, the sequence of iterates, $\{x_t\}_{t\geq 0}$ of compressed GD updates satisfy*

$$E_{C_t}(F_{t+1}) - F_\star \leq \left(1 - \frac{\delta_t \mu}{L}\right)(F_t - F_\star). \tag{10}$$

*Proof.* Using the $L$-smoothness of $F$ as in Assumption 1 we have

$$
\begin{aligned}
F_{t+1} \quad &\leq \quad F_t + \langle \nabla F_t, x_{t+1} - x_t \rangle + \frac{L}{2}\|x_{t+1} - x_t\|^2 \\
&\overset{\text{By } equation\ 2}{\leq} \quad F_t - \eta \langle \nabla F_t, \mathcal{C}_t(\nabla F(x_t)) \rangle + \frac{\eta^2 L}{2}\|\mathcal{C}_t(\nabla F(x_t))\|^2.
\end{aligned}
$$

By taking the expectation on the randomness of $\mathcal{C}_t$ and by using the GD updates from equation 2 we have

$$
\begin{aligned}
\mathbb{E}_{\mathcal{C}_t}(F_{t+1}) \quad &\leq \quad F_t - \frac{1}{L}\|\nabla F_t\|^2 + \frac{1}{2L}\mathbb{E}_{\mathcal{C}_t}\|\mathcal{C}_t(\nabla(F_t))\|^2 \\
&\overset{\text{By Lemma4}}{\leq} \quad F_t - \left(\frac{1}{L} - \frac{2 - \delta_t}{2L}\right)\|\nabla F_t\|^2 \\
&\leq \quad F_t - \frac{\delta_t}{2L}\|\nabla F_t\|^2 \\
&\overset{\text{By Assumption4}}{\leq} \quad F_t - \frac{\delta_t}{2L}2\mu(F_t - F_\star).
\end{aligned}
$$

Finally, subtracting $F_\star$ from both sides we get

$$\mathbb{E}_{\mathcal{C}_t}(F_{t+1}) - F_\star \leq \left(1 - \frac{\delta_t}{L}\mu\right)(F_t - F_\star).$$

This completes the proof. $\qquad\qquad\square$

### A.3 Convergence proofs for nonconvex distributed SGD

In this section, we provide the convergence proofs of compressed distributed SGD on nonconvex functions. We start with the key inequalities used in our proofs.

**Lemma 3.** *(Compression variance) Let $\mathcal{C}_t$ be a $\delta_t$-compressor for all $t \in [T]$, and let $F$ follow Assumption 6, and the stochastic noise follow Assumption 5. Then we have*

$$\mathbb{E}\left[\left\|\frac{1}{n}\left(\sum_{i=1}^{n}\mathcal{C}_t(g_{i,t}) - \sum_{i=1}^{n}\nabla f_{i,t}\right)\right\|^2 |x_t\right] \leq \tag{11}$$

$$\frac{1}{n}\big((1 - \delta_t)(M+1) + M\big)\big(2A(F_t - F_\star) + B + \|\nabla F_t\|^2\big) + \frac{(2 - \delta_t)\sigma^2}{n}.$$

*Proof.* We note that the compression operator, $\mathcal{C}_t$, and the stochastic noise, $\xi_{i,t}$ are independent of each other. Therefore, while taking expectation on the randomness of the compression operator, $\mathcal{C}_t$ we condition on the other source of randomness, and vice versa. We use $\mathbb{E}_{\mathcal{C}_t}$ to denote the expectation taken on the randomness of the compression operator, $\mathcal{C}_t$, and conditioned on other sources of randomness. So, taking expectation on the randomness of the compression operator, $\mathcal{C}_t$

we have

$$\mathbb{E}_{\mathcal{C}_t} \left\| \frac{1}{n} \left( \sum_{i=1}^{n} \mathcal{C}_t(g_{i,t}) - \sum_{i=1}^{n} \nabla f_{i,t} \right) \right\|^2$$

$$\overset{\mathbb{E}_{\mathcal{C}_t}(\mathcal{C}_t(g_{i,t}))=g_{i,t}}{=} \frac{1}{n^2} \sum_{i=1}^{n} \mathbb{E}_{\mathcal{C}_t} \|\mathcal{C}_t(g_{i,t}) - \nabla f_{i,t}\|^2 + \frac{2}{n^2} \sum_{i \neq j} \langle g_{i,t} - \nabla f_{i,t}, g_{j,t} - \nabla f_{j,t} \rangle$$

$$\overset{g_{i,t}=\nabla f_{i,t}+\xi_{i,t}}{=} \frac{1}{n^2} \sum_{i=1}^{n} \mathbb{E}_{\mathcal{C}_t} \|\mathcal{C}_t(g_{i,t}) - g_{i,t} + \xi_{i,t}\|^2 + \frac{2}{n^2} \sum_{i \neq j} \langle g_{i,t} - \nabla f_{i,t}, g_{j,t} - \nabla f_{j,t} \rangle$$

$$\overset{\mathbb{E}_{\mathcal{C}_t}(\mathcal{C}_t(g_{i,t}))=g_{i,t}}{=} \frac{1}{n^2} \sum_{i=1}^{n} \left( \mathbb{E}_{\mathcal{C}_t} \|\mathcal{C}_t(g_{i,t}) - g_{i,t}\|^2 + \mathbb{E}_{\mathcal{C}_t} \|\xi_{i,t}\|^2 \right) + \frac{2}{n^2} \sum_{i \neq j} \langle g_{i,t} - \nabla f_{i,t}, g_{j,t} - \nabla f_{j,t} \rangle$$

$$\leq \frac{1}{n^2} \sum_{i=1}^{n} \left( (1 - \delta_t) \|g_{i,t}\|^2 + \|\xi_{i,t}\|^2 \right) + \frac{2}{n^2} \sum_{i \neq j} \langle g_{i,t} - \nabla f_{i,t}, g_{j,t} - \nabla f_{j,t} \rangle.$$

Taking expectation conditioned on $x_t$, and by using the tower property of expectation we get

$$\mathbb{E} \left[ \mathbb{E}_{\mathcal{C}_t} \left[ \left\| \frac{1}{n} \left( \sum_{i=1}^{n} \mathcal{C}_t(g_{i,t}) - \sum_{i=1}^{n} \nabla f_{i,t} \right) \right\|^2 \right] | x_t \right] \leq \frac{1}{n^2} \sum_{i=1}^{n} \left( (1 - \delta_t) \mathbb{E}[\|g_{i,t}\|^2 | x_t] + \mathbb{E}[\|\xi_{i,t}\|^2 | x_t] \right).$$

The equality holds as $\mathbb{E}(g_{i,t}|x_t) = \nabla f_{i,t}$ and $\mathbb{E}(g_{j,t}|x_t) = \nabla f_{j,t}$, for all $i \neq j, i, j \in [n]$. By using Assumption 5, write the above expression as

$$\frac{1}{n^2} \sum_{i=1}^{n} \left( (1 - \delta_t) \mathbb{E}[\|g_{i,t}\|^2 | x_t] + \mathbb{E}[\|\xi_{i,t}\|^2 | x_t] \right)$$

$$\leq \frac{1}{n^2} \sum_{i=1}^{n} \left( (1 - \delta_t)(M + 1) \|\nabla f_{i,t}\|^2 + (1 - \delta_t)\sigma^2 + M \|\nabla f_{i,t}\|^2 + \sigma^2 \right)$$

$$\overset{\text{By Lemma 5}}{\leq} \frac{1}{n} \left( (1 - \delta_t)(M + 1) + M \right) \left( 2A(F_t - F_\star) + B + \|\nabla F_t\|^2 \right) + \frac{1}{n}(2 - \delta_t)\sigma^2.$$

Hence the result. $\qquad \square$

Based on the previous Lemma, the next lemma quantifies the quantity $\mathbb{E} \left\| \frac{1}{n} \sum_{i=1}^{n} \mathcal{C}_t(g_{i,t}) \right\|^2$.

**Lemma 6.** *Let $\mathcal{C}_t$ be a $\delta_t$-compressor for all $t \in [T]$. Let $F$ follow Assumptions 3, 6, and the stochastic noise follow Assumption 5. Then*

$$\mathbb{E} \left\| \frac{1}{n} \sum_{i=1}^{n} \mathcal{C}_t(g_{i,t}) \right\|^2 \leq \frac{2A\beta_t}{n}(F_t - F_\star) + \left( 1 + \frac{\beta_t}{n} \right) \|\nabla F_t\|^2 + \frac{B\beta_t}{n} + \left( \frac{2 - \delta_t}{n} \right) \sigma^2, \tag{12}$$

*where $\beta_t := (1 - \delta_t)(M + 1) + M$.*

*Proof.* Taking expectation on the randomness of the compression operator, $\mathcal{C}_t$, we have

$$\mathbb{E}_{\mathcal{C}_t} \left\| \frac{1}{n} \sum_{i=1}^{n} \mathcal{C}_t(g_{i,t}) \right\|^2 = \mathbb{E}_{\mathcal{C}_t} \| \frac{1}{n} \sum_{i=1}^{n} \mathcal{C}_t(g_{i,t}) - \nabla F_t + \nabla F_t \|^2$$

$$= \mathbb{E}_{\mathcal{C}_t} \left\| \frac{1}{n} \sum_{i=1}^{n} \mathcal{C}_t(g_{i,t}) - \nabla F_t \right\|^2 + \|\nabla F_t\|^2 + 2 \langle \frac{1}{n} \sum_{i=1}^{n} g_{i,t} - \nabla F_t, \nabla F_t \rangle$$

$$\overset{\text{By Lemma 3}}{\leq} \frac{1}{n} \left( (1 - \delta_t)(M + 1) + M \right) \left( 2A(F_t - F_\star) + B + \|\nabla F_t\|^2 \right) + \frac{1}{n}(2 - \delta_t)\sigma^2$$

$$+ \|\nabla F_t\|^2 + 2 \langle \frac{1}{n} \sum_{i=1}^{n} g_{i,t} - \nabla F_t, \nabla F_t \rangle. \tag{13}$$

Finally, we note that $\mathbb{E}(g_{i,t}|x_t) = f_{i,t}$. By using the tower property of expectation, we denote $\mathbb{E}\|\frac{1}{n}\sum_{i=1}^n \mathcal{C}_t(g_{i,t})\|^2 = \mathbb{E}(\mathbb{E}_{\mathcal{C}_t}\|\frac{1}{n}\sum_{i=1}^n \mathcal{C}_t(g_{i,t})\|^2|x_t)$. Taken together, from 13, we have

$$\mathbb{E}\|\frac{1}{n}\sum_{i=1}^n \mathcal{C}_t(g_{i,t})\|^2$$

$$\leq \quad \frac{1}{n}\left((1-\delta_t)(M+1)+M\right)\left(2A(F_t - F_\star)+B+\|\nabla F_t\|^2\right) + \frac{1}{n}(2-\delta_t)\sigma^2 + \|\nabla F_t\|^2.$$

Hence the result. $\qquad\square$

Finally, we can quote the non-convex descent lemma for compressed distributed SGD.

**Lemma 7.** *(Non-convex descent lemma) Let Assumptions 1 , 5, and 6 hold, and let $\mathcal{C}_t$ be a $\delta_t$-compressor for all $t \in [T]$. Then*

$$\mathbb{E}(F_{t+1}) - F_\star \quad \leq \quad \left(1 + \frac{AL\eta_t^2\beta_t}{n}\right)(\mathbb{E}(F_t) - F_\star) - \eta_t\left(1 - \frac{L\eta_t}{2} - \frac{L\eta_t\beta_t}{n}\right)\mathbb{E}\|\nabla F_t\|^2$$

$$+ \frac{L\eta_t^2}{2}\left(\frac{B\beta_t}{n} + \left(\frac{2-\delta_t}{n}\right)\sigma^2\right).$$

*Proof.* By using the $L$-smoothness of $F$ we have

$$F_{t+1}\leq \quad F_t - \langle\nabla F_t, x_{t+1} - x_t\rangle + \frac{L}{2}\|x_{t+1} - x_t\|^2.$$

By using the update rule $x_{t+1} - x_t = -\frac{\eta_t}{n}\sum_{i=1}^n \mathcal{C}_t(g_{i,t})$ the above becomes

$$F_{t+1}\leq \quad F_t - \langle\nabla F_t, \frac{\eta_t}{n}\sum_{i=1}^n \mathcal{C}_t(g_{i,t})\rangle + \frac{L\eta_t^2}{2}\|\frac{1}{n}\sum_{i=1}^n \mathcal{C}_t(g_{i,t})\|^2. \qquad (14)$$

Taking expectation with respect to the randomness of $\mathcal{C}_t$ on the above expression for all $t \in [T]$, we find

$$\mathbb{E}_{\mathcal{C}_t}(F_{t+1})\leq \quad F_t - \langle\nabla F_t, \frac{\eta_t}{n}\sum_{i=1}^n g_{i,t}\rangle + \frac{L\eta_t^2}{2}\mathbb{E}_{\mathcal{C}_t}\|\frac{1}{n}\sum_{i=1}^n \mathcal{C}_t(g_{i,t})\|^2.$$

Taking expectation conditioned on $x_t$ we have

$$\mathbb{E}(F_{t+1}|x_t) \quad \leq \quad \mathbb{E}(F_t|x_t) - \eta_t\mathbb{E}\|\nabla F_t\|^2 + \frac{L\eta_t^2}{2}\mathbb{E}\left(\|\frac{1}{n}\sum_{i=1}^n \mathcal{C}_t(g_{i,t})\|^2|x_t\right).$$

By using Lemma 6 on the above we find

$$\mathbb{E}(F_{t+1}|x_t) \quad \leq \quad \mathbb{E}(F_t|x_t) - \eta_t\mathbb{E}\|\nabla F_t\|^2$$

$$+ \frac{L\eta_t^2}{2}\left(\frac{2A\beta_t}{n}(F_t - F_\star) + \left(1 + \frac{\beta_t}{n}\right)\|\nabla F_t\|^2 + \frac{B\beta_t}{n} + \left(\frac{2-\delta_t}{n}\right)\sigma^2\right).$$

Taking the final expectation, by using the tower property of expectation, and rearranging the terms, we have

$$\mathbb{E}(F_{t+1}) - F_\star \quad \leq \quad \left(1 + \frac{AL\eta_t^2\beta_t}{n}\right)(\mathbb{E}(F_t) - F_\star) - \eta_t\left(1 - \frac{L\eta_t}{2} - \frac{L\eta_t\beta_t}{n}\right)\mathbb{E}\|\nabla F_t\|^2$$

$$+ \frac{L\eta_t^2}{2}\left(\frac{B\beta_t}{n} + \left(\frac{2-\delta_t}{n}\right)\sigma^2\right). \qquad (15)$$

Hence the result.

$\qquad\square$

NONCONVEX CONVERGENCE RESULTS

The next Lemma is instrumental in proving the nonconvex convergence of distributed SGD with $\delta$-compressors.

**Lemma 8.** *Mishchenko et al. (2020) Let for $0 \leq t \leq T$ the following holds:*

$$p_{t+1} \leq (1+a)p_t - bq_t + c, \tag{16}$$

*where $\{p_t\}_{t=0}^{T}$ and $\{q_t\}_{t=0}^{T}$ are non-negative sequences and $a, b, c \geq 0$ are constants. Then*

$$\min_{t=0,1,\cdots T-1} q_t \leq \frac{(1+a)^T}{bT} p_0 + \frac{c}{b}. \tag{17}$$

*Proof.* Dividing both sides of equation 16 by $(1+a)^{t+1}$ and summing from $t = 0, 1, \cdots, T$ we have

$$\sum_{t=0}^{T} \frac{1}{(1+a)^{t+1}} p_{t+1} \leq \sum_{t=0}^{T} \frac{1}{(1+a)^t} p_t - \sum_{t=0}^{T} \frac{b}{(1+a)^{t+1}} q_t + \sum_{t=0}^{T} \frac{c}{(1+a)^{t+1}},$$

which after rearranging is

$$\sum_{t=0}^{t} \frac{b}{(1+a)^{t+1}} q_t \leq p_0 - \frac{1}{(1+a)^{T+1}} p_{T+1} + \sum_{t=0}^{T} \frac{c}{(1+a)^{t+1}}.$$

Noting $\sum_{t=0}^{T} \frac{1}{(1+a)^{t+1}} \leq \frac{1}{1-\frac{1}{1+a}} - 1 = \frac{1}{a}$, we have

$$\min_{t=0,1,\cdots T} q_t \sum_{t=0}^{T} \frac{1}{(1+a)^{t+1}} \leq \sum_{t=0}^{T} \frac{1}{(1+a)^{t+1}} q_t \leq \frac{p_0}{b} + \frac{c}{ab}. \tag{18}$$

Hence the result. $\qquad\square$

Finally, we are all set to prove Theorem 1.

**Theorem 1.** *(Nonconvex convergence) Let Assumptions 1 , 5, and 6 hold, and let $\mathcal{C}_t$ be a $\delta_t$-compressor for all $t \in [T]$. For a fixed stepsize $\eta_t := \eta \leq \min\left(\frac{1}{\frac{L}{2} + \frac{L(2M+1)}{n}}, \left(\frac{AL(2M+1)T}{n}\right)^{-\frac{1}{2}}\right)$ we have:*

$$\min_{t=0,1,\cdots T-1} \mathbb{E}\|\nabla F(x_t)\|^2 \leq \frac{3}{T\eta\left(1 - \frac{L\eta}{2} - \frac{L\eta}{n}\right)} (F_0 - F_\star) + \frac{L\eta\left(B(2M+1) + 2\sigma^2\right)}{2n\left(1 - \frac{L\eta}{2} - \frac{L\eta(2M+1)}{n}\right)}.$$

*Proof.* From Lemma 7 we have

$$\begin{aligned}
\mathbb{E}(F_{t+1}) - F_\star &\leq \left(1 + \frac{AL\eta_t^2 \beta_t}{n}\right)(\mathbb{E}(F_t) - F_\star) - \eta_t \left(1 - \frac{L\eta_t}{2} - \frac{L\eta_t \beta_t}{n}\right) \mathbb{E}\|\nabla F_t\|^2 \\
&\quad + \frac{L\eta_t^2}{2}\left(\frac{B\beta_t}{n} + \left(\frac{2-\delta_t}{n}\right)\sigma^2\right).
\end{aligned}$$

The above inequality satisfies the condition of equation 16 with $a = \frac{AL\eta^2(2M+1)}{n}, b = \eta\left(1 - \frac{L\eta}{2} - \frac{L\eta(2M+1)}{n}\right), c = \frac{L\eta^2}{2}\left(\frac{B(2M+1)}{n} + \frac{2\sigma^2}{n}\right)$. Therefore, we obtain

$$\min_{t=0,1,\cdots T-1} \mathbb{E}\|\nabla F(x_t)\|^2 \leq \frac{\left(1 + \frac{AL\eta^2(2M+1)}{n}\right)^T}{T\eta\left(1 - \frac{L\eta}{2} - \frac{L\eta(2M+1)}{n}\right)} (F_0 - F_\star) + \frac{\frac{L\eta^2}{2}\left(\frac{B(2M+1)}{n} + \frac{2\sigma^2}{n}\right)}{\eta\left(1 - \frac{L\eta}{2} - \frac{L\eta(2M+1)}{n}\right)}. \tag{19}$$

Using that $x+1 \leq \exp x$ and with $\eta \leq \left(\frac{AL(2M+1)T}{n}\right)^{-\frac{1}{2}}$ in the first term of the RHS of equation 19, we get

$$\left(1 + \frac{AL\eta^2(2M+1)}{n}\right)^T \leq \exp\left(\frac{AL\eta^2(2M+1)T}{n}\right) \leq \exp(1) \leq 3.$$

Finally, using the above in the the inequality (19), we have

$$\min_{t=0,1,\cdots T-1} \mathbb{E}\|\nabla F(x_t)\|^2 \leq \frac{3}{T\eta\left(1 - \frac{L\eta}{2} - \frac{L\eta}{n}\right)} (F_0 - F_\star) + \frac{L\eta\left(B(2M+1) + 2\sigma^2\right)}{2n\left(1 - \frac{L\eta}{2} - \frac{L\eta(2M+1)}{n}\right)}.$$

Hence the result.

$\qquad\square$

## B    Supplementary Numerical Results

In this section, we provide additional experimental details and benchmarking results, which we were unable to discuss in the main paper due to limited space.

Table 2: Summary of the benchmarks and quality metrics used in this work.

| Task | Model | Dataset | Training parameters | Quality metric | Baseline quality | Optimizer |
|------|-------|---------|---------------------|----------------|------------------|-----------|
| Image Classification | AlexNet | CIFAR-10 | 2,255,296 | Accuracy | 84.99% | SGD Robbins & Monro (1951) |
| | ResNet9 | CIFAR-10 | 6,573,120 | Accuracy | 92.07% | SGD Robbins & Monro (1951) |
| | ResNet18 | CIFAR-100 | 11,220,132 | Accuracy | 73.43% | SGD-M Nesterov (2013) |
| | ResNet50 | ImageNet | 25,559,081 | Accuracy | 59.43% | SGD Robbins & Monro (1951) |
| Recommendation | NCF | Movielens-20m | 31,832,577 | HR@10 | 95.53% | ADAM Kingma & Ba (2015) |
| Language Modelling | Transformer-XL | WikiText-103 | 191,950,298 | Perplexity | 39.47 | LAMB You et al. (2020) |
| Federated Learning | ResNet18 | CIFAR-10 | 11,173,962 | Accuracy | 85.37% | SGD-M Nesterov (2013) |

### B.1    Performance of Random-$k$ in LEGACY as base compressor: Accuracy vs. Data volume

We provide additional tests following the configuration described in Section 5, using the Random-$k$ as the base compressor in LEGACY. Figure 7 displays the accuracy versus relative average data volume throughout training for AlexNet, ResNet-9, and Transformer-XL.

### B.2    Average of Independent Runs

In Table 4, we report the accuracy of ResNet-9 and AlexNet, including standard deviations obtained through independent runs using Top-$k$ and Random-$k$ as base compressors in LEGACY. Top-$k$ demonstrates superior performance relative to Random-$k$. The tests conducted reveal comparable findings to those discussed in Subsection 5.1, further validating the importance of small layers and the initial training phase in improving compression efficiency.

### B.3    Model quality vs. run time

We performed our previous experiments on high-performance GPUs in a data center, connected by a fast network, and constituting a limited number of workers. To simulate more constrained environments, we now simulate scenarios with more restricted resources.

**Testbed and setup.** We trained ResNet-18 on CIFAR10 using 50 workers, sharing a 1Gbps network bandwidth, with every worker operating on an Intel Xeon Platinum 8276 CPU instead of a GPU. In this part, we integrated error feedback (EF) in our tests; the implementation of EF is based on Sahu et al. (2021). Figure 8 profiles the accuracy per wall clock time for 4100 seconds, which is the time required for compressors to complete 30 epochs. For the compression parameters of each method, we employed the following so that all methods transmit (almost) equal average data volume:

Table 3: Dataset and training configuration.

| Dataset Name | Size | Workers used | Training Time (min) | Independent Runs Performed |
|--------------|------|--------------|---------------------|----------------------------|
| CIFAR10 Krizhevsky et al. (2009) | 160MB | 2 | 5 | 15 |
| CIFAR100 Krizhevsky et al. (2009) | 160MB | 2 | 20 | 15 |
| ImageNet Deng et al. (2009) | 140GB | 4 | 2100 | 1 |
| Movielens-20m Harper & Konstan (2015) | 190MB | 4 | 2 | 10 |
| WikiText-103 Merity et al. (2017) | 500MB | 4 | 190 | 4 |

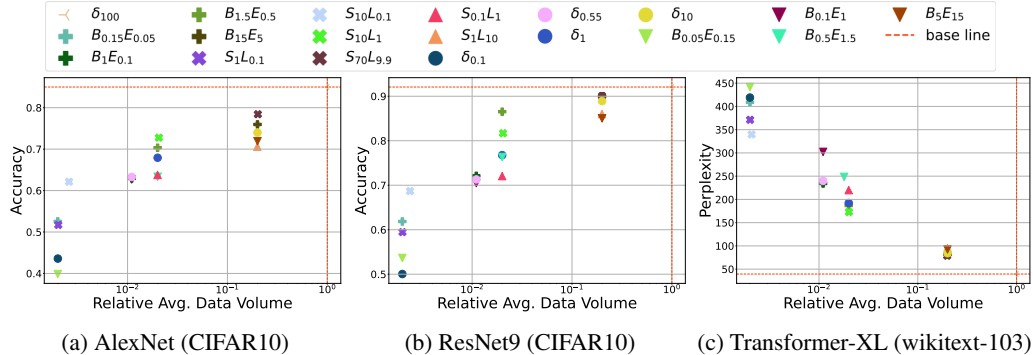

| (a) AlexNet (CIFAR10) | (b) ResNet9 (CIFAR10) | (c) Transformer-XL (wikitext-103) |

Figure 7: Layer-size and training epoch dependent Random-$k$ compression, where $S_{\delta_1}L_{\delta_2}$ means small layers($\leq 10^5$) compressed with compression factor, $\delta_1$ and large layers compressed with compression factor, $\delta_2$, and $B_{\delta_1}E_{\delta_2}$ denotes two-phase training, beginning phase (half of the total training epoch) with compression factor, $\delta_1$ and ending phase with compression factor $\delta_2$.

Table 4: Comparison of average compression ratios vs. mean accuracy with standard deviation derived from 7 runs.

| | | ResNet9 | | AlexNet | |
| Method | Compression ratio | Average ratio | Accuracy | Average ratio | Accuracy |
| --- | --- | --- | --- | --- | --- |
| Baseline | N/A | 100% | $92.07 \pm 0.13$ | 100% | $84.98 \pm 0.34$ |
| Topk | 0.1% | 0.1% | $75.72 \pm 1.07$ | 0.1% | $65.53 \pm 0.86$ |
| Topk-epoch | $B_{0.05}E_{0.15}$ | 0.1% | $73.65 \pm 0.16$ | 0.1% | $59.85 \pm 4.9$ |
| Topk-epoch | $B_{0.15}E_{0.05}$ | 0.1% | $79.18 \pm 0.26$ | 0.1% | $66.25 \pm 0.62$ |
| Topk-layer | $S_{10}L0.1$ | 0.12% | $\mathbf{82.94 \pm 0.79}$ | 0.13% | $\mathbf{70.27 \pm 0.91}$ |
| Randomk | 0.1% | 0.1% | $50.04 \pm 0.8$ | 0.1% | $43.58 \pm 0.45$ |
| Randomk-layer | $S_{10}L_{0.1}$ | 0.12% | $\mathbf{68.67 \pm 0.53}$ | 0.13% | $\mathbf{62.13 \pm 0.45}$ |

- Top-$k$: 1.7% uniform compression.

- Accordion: Set low and high compression ratio to $k_{low} = 0.1\%$ and $k_{high} = 10\%$, respectively, achieving an average compression ratio of 1.98%.

- Top-$k$ Epoch-based: The total training duration of 30 epochs was divided into four segments: three segments of 8 epochs each, followed by a final segment of 7 epochs. Compression ratios were set to 5%, 1%, 0.5%, and 0.1% for each segment, respectively, resulting in an average compression ratio of 1.75%.

- Top-$k$ Layer-based: Layers were categorized based on size into five groups: very small ($\leq$ 100), small ($\leq$ 600), medium ($\leq 10^5$), large ($\leq 10^6$), and very large ($\geq 10^6$). Assigned compression ratios were 80%, 50%, 20%, 5%, and 0.1% for each group respectively, transmitting 1.77% of the gradients.

**Results.** Although the no-compression baseline achieves the highest accuracy, the time required is also large in environments with limited and weak resources. In this test, the baseline needed more than 6 hours to complete 30 epochs, while the compression tests took $\approx$ 4100 seconds, thereby achieving the best return for time. From Figure 8a, we can observe that the Epoch-based Top-$k$ strategy achieves the best performance in the first 1000 seconds, which is expected as the method is running through a light compression of 5% during this period, compared to the other compressors that are using around a 1.7% compression ratio. The uniform compressors required approximately double the time ($\approx 2000s$) to reach this level of accuracy. On the other hand, the Top-$k$ strategy based on layer size, stands out with the best accuracy when the layer size groups are more refined, creating more groups helps in controlling the compression for sensitive and small layers to achieve better accuracy.

**Takeaways.** In resource-limited environments, the strategies in `LEGACY` perform better in terms of obtaining a better accuracy faster. The initial mild compression phase of the epoch-based strategy

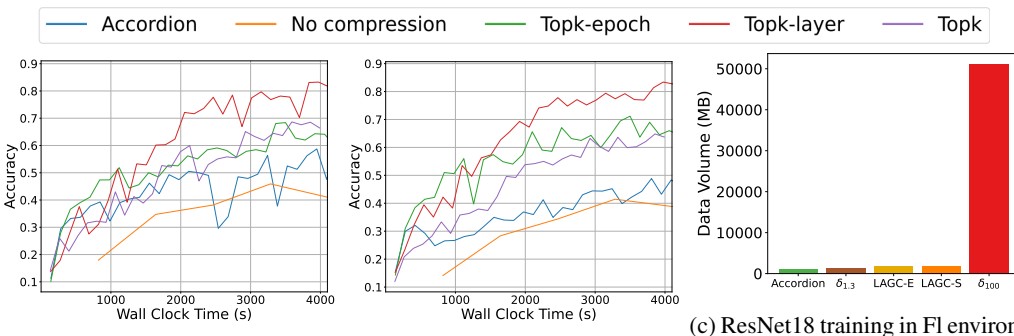

(a) ResNet18 training (with EF)   (b) ResNet18 training (without EF)   (c) ResNet18 training in Fl environ-
ment, total data volume.

Figure 8: In (a) and (b), we show accuracy vs. wall clock time of training ResNet-18 on CIFAR10, with and without EF, respectively. In (c), we show the total communicated data volume in ResNet-18 on CIFAR10 training in an FL environment; see legend in Figure 6.

allows it to benefit from the early training phase and outperform other methods, which take significant time to match its performance, even after the epoch strategy enters the aggressive phase. On the other hand, applying light compression to small layers enhances model performance. In both strategies, creating more groups aids in refining the compression more effectively to achieve better performance.

## B.4 TIME COMPLEXITY

The time complexity of `LEGACY` is equivalent to the time complexity of the base compressor used in it. `LEGACY` does not involve any back-of-the-hand calculation in choosing the adaptive version of the compressor.

## B.5 REPRODUCIBILITY

We implement the sparsifiers in `PyTorch`. Tables 5, 6, 7, 8, and 9 provide the experimental details for each of the tasks. We used the default hyperparameters provided in the mentioned repositories for each task.

Table 5: CIFAR-10 experiments

| Dataset | CIFAR-10 |
|---|---|
| Architecture | AlexNet, ResNet-9 |
| Repository | Layer-Wise-AAAI20 Dutta et al. (2020) |
| | See `https://github.com/sands-lab/layer-wise-aaai20` |
| License | MIT |
| Number of workers | 2 |
| Global Batch-size | $256 \times 2$ |
| Optimizer | vanilla SGD |
| LR scheduler | piecewise-linear function that increases the |
| | learning rate from 0 to 0.4 during the first 5 epochs |
| | and then decreases to 0 till the last epoch |
| Number of Epochs | 30 |
| Repetitions | 15, with different seeds |

## C LIMITATION, FUTURE DIRECTION, AND ETHICS STATEMENT

Although adapting compression ratios according to layer size and training phase can significantly improve model performance, it requires an additional set of hyperparameters. These parameters determine the number of layer groups to create, when to adjust compression levels (start new phase),

Table 6: CIFAR-100 experiments

| | |
|---|---|
| Dataset | CIFAR-100 |
| Architecture | ResNet-18 |
| Repository | rethinking-sparsification Sahu et al. (2021) |
| | See `https://github.com/sands-lab/rethinking-sparsification` |
| License | MIT |
| Number of workers | 2 |
| Global Batch-size | $256 \times 2$ |
| Optimizer | SGD with Nesterov Momentum |
| Momentum | 0.9 |
| Post warmup LR | $0.1 \times 16$ |
| LR-decay | /10 at epoch 150 and 250 |
| LR-warmup | Linearly within 5 epochs, starting from 0.1 |
| Number of Epochs | 300 |
| Weight decay | $10^{-4}$ |
| Repetitions | 15, with different seeds |

Table 7: Language modelling task

| | |
|---|---|
| Dataset | WikiText103 |
| Architecture | Transformer-XL |
| Repository | NVIDIA Deep Learning Examples Nvidia |
| | See `https://github.com/NVIDIA/DeepLearningExamples` |
| License | Apache |
| Number of workers | 4 |
| Global Batch-size | 256 |
| Optimizer | LAMB |
| LR-decay | Cosine schedule from 0.01 to 0.001 |
| LR-warmup | Linearly within 1,000 iterations, reaching 0.01 |
| Number of training steps | 4500 |
| Weight decay | 0 |
| Repetitions | 4, with different seeds |

Table 8: Recommendation task

| | |
|---|---|
| Dataset | Movielens-20M |
| Architecture | NCF |
| Repository | NVIDIA Deep Learning Examples Nvidia |
| | See `https://github.com/NVIDIA/DeepLearningExamples` |
| Number of workers | 2 |
| Global Batch-size | $2^{20}$ |
| Optimizer | ADAM |
| ADAM $\beta_1$ | 0.25 |
| ADAM $\beta_2$ | 0.5 |
| ADAM LR | $4.5 \times 10^{-3}$ |
| Number of Epochs | 30 |
| Weight decay | 0 |
| Dropout | 0.5 |
| Repetitions | 10, with different seeds |
| License | Apache |

Table 9: ImageNet experiments

| Dataset | ImageNet |
|---|---|
| Architecture | ResNet-50 |
| Repository | PyTorch Examples PyTorch |
| | See `https://github.com/pytorch/examples` |
| License | BSD 3-Clause |
| Number of workers | 4 |
| Global Batch-size | 256 |
| Optimizer | SGD |
| Momentum | 0.9 |
| LR-decay | LR decayed by 10 every 30 epochs |
| Number of Epochs | 50 |
| Weight decay | $10^{-4}$ |
| Repetitions | 1 |

and which compression ratios to apply for each group during each training phase. We note that choosing these hyperparameters does not require any rigorous setup compared with other state-of-the-art adaptive compressors. Regardless, in the following, we discuss some test cases. In the future, we plan to make these choices more robust.

In our experiments, we followed a simple approach for grouping layers that involves sorting the model's layers by their sizes and identifying any significant differences to establish new groups. Determining optimal compression ratios for each group is less straightforward but can be managed by incrementally adjusting the aggressiveness for larger layers and redistributing the gain among other groups. E.g., shifting from a uniform 10% to using $S_{70}L_{9.9}$ achieves similar average compression ratios: 10.075%, 10.003%, 10.032%, 9.917%, and 9.24% for AlexNet, ResNet9, ResNet18, NCF, and Transformer-XL, respectively. Deciding when to start a new training phase is also challenging. For simplicity, one can evenly divide the total number of training epochs into the desired number of phases.

We run some proof of concept experiments using error feedback, or memory compensation. However, investigating the effect of error feedback further into our `LEGACY` is non-trivial both empirically and theoretically, and is left for future work.

**Potential Negative impact.** Gradient compression techniques have been widely adopted since their introduction to the machine learning community. The strategies used in developing our adaptive compression scheduler in this work theoretically and empirically demonstrate their capability of achieving better accuracy in DNN training in a distributed and federated setup. Overall, the present work is theoretically driven and experiments corroborate the theoretical claims. Therefore, we do not find any foreseeable harm it can pose to human society. However, it is always possible that some individual or an organization can use this idea to devise a *technique* that can appear harmful to society and bear evil consequences. As authors, we are absolutely against any detrimental usage, regardless, by any individual or an organization, under profit or non-profitable motivation, and pledge not to support any detrimental endeavors concerning our idea therein.

