# OpenReview forum: "LEGACY: A Lightweight Adaptive Gradient Compression Strategy for Distributed Deep Learning"
_ICLR.cc/2025/Conference — ICLR 2025 Conference Withdrawn Submission_

### Official Review · Reviewer_9E5h · 2024-11-03

**Soundness:** 3
**Presentation:** 3
**Contribution:** 3
**Rating:** 5
**Confidence:** 3

**Summary:**

The paper focuses on the problem of gradient compression in distributed learning. The main contribution is an adaptive scheduler for any compressors to guide the compression parameters selection. By taking into account the layer size of the DNNs and the training phases, the schedule is able to reduce communication overhead in an adaptive fashion. The approach can work with any compression technique. Experimental evaluation is performed on different models and datasets and shows very good results.

**Strengths:**

The paper is very clearly written and gives an excellent review of related work in the field. It very clearly states where its main contributions lie and back up the methodological contributions by theoretical analyses. I very much like the structure of the paper and the incorporation of the theoretical analysis.
The paper is original in the sense that it targets a very specific question, newly training epoch and layer size dependency in distributed training. Although the experimental evaluation is performed on different datasets and models, it is difficult to say how generalisable the proposed approach really is and how impactful it is in practice.

**Weaknesses:**

The approach focuses on two aspects, namely training epoch and layer size dependency. Are these the only two aspects which are important or are there more aspects, which are not included but which are relevant in practice?
The practical impact of the proposed approach is not 100% clear. What do the compression results mean in practice?  What are the limitations of the approach? The method uses Top-k as the base compressor, but there are many more methods in the field. How to the approach compare to them?
Since the technical contribution of the paper is rather limited and the paper proposes mainly a (more or less) heuristic adaptive scheduling approach, the experimental evaluation could contain more in-depth analyses and also analyses showing the real practical benefits of the approach.

**Questions:**

Are other aspects beyond training epoch and layer size dependent relevant, which are not considered here?
What are the limitations of the approach?
What are the practical benefits?

---

> ### Author Response · Authors · 2024-11-20
>
> We thank the reviewer for the overall positive evaluation of our work. Below we answer the questions raised by the reviewer.
>
> **Investigating other factors in LEGACY.** There are other factors that are interesting to investigate such as the layers type. In the main paper, we focused on the two factors for which we have theoretical insights and numerical evidence. Exploring other factors is left for future work.
>
> **Question.** The practical impact of the proposed approach is not 100% clear. What do the compression results mean in practice?
>
>  **Answer.** As digital datasets are getting bigger, modern DNN models are getting more intricate to find proper decisions from the data, and one of the fundamental challenges in training these models is their increasing sizes. For efficient training, practitioners widely employ distributed parallel training over multiple computing nodes/workers where in each iteration, gradient from each worker is exchanged synchronously through the network for aggregation, and the aggregated global gradient is sent back to the workers. The workers jointly update the model parameters by using the global gradient. However, the network latency during gradient transmission creates a communication bottleneck and as a result, slows down the training. To remedy this, different gradient compression techniques are proposed to reduce the communicated data volume and facilitate faster training. For a detailed overview and practical impact of compression techniques see [Grace: A compressed communication framework for distributed machine learning, Xu et al. In IEEE ICDCS, 2021] and the references therein.
>
> **Question.** What are the limitations of the approach?
>
> **Answer.** Please see **LIMITATION, FUTURE DIRECTION, AND ETHICS STATEMENT** in the Appendix (**Section C**, pp.24). We mentioned this explicitly in the main paper; see Section 5.3 **ADDITIONAL BENCHMARKING AND DISCUSSIONS**, Line-473.
>
> **Question.** The method uses Top-k as the base compressor, but there are many more methods in the field. How to the approach compare to them?
>
> **Answer.** We compared our approach with several approaches from the SOTA in the paper. As per the combined compression that we use. We also conducted experiments using random-k; see the Appendix. During rebuttal, we embedded two more popular compression techniques: random dithering, also, popularly known as quantized SGD (QSGD), and low-rank compression technique, PowerSGD in LEGACY and compared them with base uniform compression and two other adaptive compressors; see our **Extra Experimental Results** as provided to **Reviewer-qMHq**. The strategies in LEGACY either outperform or are comparable to them.
>
> **Comment.** Since the technical contribution of the paper is rather limited and the paper proposes mainly a (more or less) heuristic adaptive scheduling approach, the experimental evaluation could contain more in-depth analyses and also analyses showing the real practical benefits of the approach.
>
>  **Response to the comment.** We disagree with the reviewer. We benchmark LEGACY through a variety of numerical experiments involving diverse DNN architectures **(convolution and residual networks, transformer, and recommender system—a total of 6 models)** trained for different tasks **(image classification on CIFAR10, CIFAR 100, and ImageNet, text prediction on WikiText-103, and collaborative filtering on Movielens-20M —a total of 5 datasets)** by using Top-k and Random-k as base compressors. Additionally, we emulate a real-world bandwidth-limited federated training environment where the network bandwidth can pose a serious communication bottleneck and deploy LEGACY and compare its efficacy against 3 state-of-the-art adaptive compressors and baseline uniform compressors; see Section 5.5.
>
> We also note that our approaches did not stem from “heuristic," it was experimental observations as we provide a detailed outline in the Introduction and later, we theoretically established these observations by analyzing the impact of gradient compressors on the decrease rate for the gradient descent algorithm in Section 3.
>
> **Question.** Are other aspects beyond training epoch and layer size dependent relevant, which are not considered here?
>
> **Answer.** Our present work highlights that designing a simple adaptive counterpart of any compression technique can be based on the training phase or DNN layer sizes and is better than uniform compression. LEGACY introduces a compute-free adaptive compression approach, in contrast to the design of other adaptive compression strategies in the literature. Other factors are interesting to investigate such as the configuration of the DNN layers. In this paper, we focused on the two factors for which we have theoretical insights and numerical evidence. Exploring other factors is left for future work.

---

> > ### Author Response · Authors · 2024-11-20
> >
> > **Question.** What are the practical benefits?
> >
> > **Answer.** One of the practical benefits of compression techniques is bandwidth-constrained federated training, which trains models collaboratively on data stored in resource-constrained, heterogeneous, and geographically remote client devices, e.g., smartphones or internet-of-things (IoTs). At each round, a fraction of the participating clients is selected. After synchronizing with the server-side global model, the selected clients perform local training for some epochs over their private data. The server collects those updates, aggregates and applies them to the global model. Communication compression becomes essential in bandwidth- and resource-constrained federated training and there is a plethora of work focused on this aspect; see [1,2,3] among many.
> >
> > **References**
> >
> > 1. Advances and Open Problems in Federated Learning, Kairouz et al., 2021.
> >
> > 2. A systematic review of federated learning: Challenges, aggregation methods, and development tools, Guendouzi et al., 2023.
> >
> > 3. Advances and Open Challenges in Federated Foundation Models, Chen et al., 2024.

---

> > > ### Comment · Reviewer_9E5h · 2024-11-25
> > > **Thank you for the additional clarification**
> > >
> > > I decided to keep my rating as is.

---

> > > > ### Author Response · Authors · 2024-11-25
> > > >
> > > > Could you please let us know if you still have any concerns or comments regarding the paper, including our rebuttal? It took us over a year to develop the paper, incorporating both theoretical insights and extensive numerical evidence.

---

> > > > ### Author Response · Authors · 2024-11-28
> > > >
> > > > Based on the reviewer's questions, we seriously doubt the reviewer’s self-defined confidence level and we also doubt how much the reviewer read our paper. For example, the reviewer asked **The practical impact of the proposed approach is not 100% clear. What do the compression results mean in practice?** and **What are the practical benefits?** These are trivial questions for researchers even remotely familiar with gradient compression, which is a booming field in ML. Next, the reviewer asked **What are the limitations of the approach?** which we mentioned in the main paper and had a section dedicated to this. We remind the reviewer about the ICLR reviewer guideline about *Supplementary* that says **It is not necessary to read supplementary material but such material can often answer questions that arise while reading the main paper, so consider looking there before asking authors.**

---

### Official Review · Reviewer_KVdq · 2024-11-03

**Soundness:** 2
**Presentation:** 3
**Contribution:** 2
**Rating:** 5
**Confidence:** 3

**Summary:**

The paper introduces a novel adaptive gradient compressor named LEGACY, to improve the communication efficiency of deep neural networks in distributed training. The performance of LEGACY in different models and tasks is verified by experiment, and its advantages over existing compression methods are demonstrated. Based on the information of layer size and training stage, this study provides a framework for dynamically selecting compression parameters, thus reducing the communication overhead while ensuring model performance

**Strengths:**

The LEGACY strategy combines the dynamic adjustment of the layer size and training phase, providing a new idea for adaptive compression, and has some innovation.

This paper provides sufficient empirical data to support the effectiveness of the proposed method through experiments on a variety of DNN architectures and tasks.

LEGACY is compatible with a wide range of compression technologies and is adaptable to different DNN models and data sets

**Weaknesses:**

Introduction part seems to describe to much details about the experiment. Though I understand that the author wants to show the finds of two key factors, it’s better to write in the later section instead of the Introduction section. Overall, the organization of the whole article needs to be improved.

The author seems to have identified only two key factors and tried to apply them. But in fact, no specific algorithm has been proposed to achieve true adaptive mechanism, so the innovation point is not enough.

Although the paper introduces two key factors for adaptive compression (training phase and layer size), it lacks a rigorous theoretical foundation to support their combined effects.  Without a deeper theoretical framework, it is challenging to determine whether these factors are universally applicable across diverse model architectures and datasets.

**Questions:**

How is the list of LEGACY compression factors selected? It doesn't seem to be adaptive.

In line 323, the author give the point that “LEGACYcan be used for uplink and downlink bidirectional compression”. However, the principle of upstream compression and downstream compression is not exactly the same, how to prove that this method can be applied to downstream compression?

Whether this method can be integrated with hard threshold compressors and how it works? The selection of threshold is different among different models and it’s hard to define.

---

> ### Author Response · Authors · 2024-11-20
>
> We thank the reviewer for an overall positive review. Below we answer the questions raised by the reviewer.
>
> **Compression factor selection in LEGACY.** LEGACY’s adaptive compression strategy is based on two fundamental factors in DNN training, layer size and training phase. Larger layers can be compressed more severely, and the later training phase can tolerate more aggressive compression without affecting the model’s performance compared to the no-compression baseline. While the layer sizes can be determined and grouped based on their relative sizes, the only rule for choosing compression parameters based on the training phase is to *choose decreasing compression parameters over iterations.* In the main paper, for simplicity, we split the training into two phases and layers into two groups, small and large; see LEGACY setup in Lines 411-417. However, a more fine-grained breakdown of each factor is possible; see our **Extra Experimental Results** as provided to **Reviewer-qMHq**. One can change the compression parameters at each iteration or each epoch or after some number of iterations. The same granularity holds for layer sizes. The adaptivity of LEGACY comes from the fact that the compression ratios only depend on the training phase or layer sizes that can be used with any compressor to make their adaptive counterpart.
>
> **LEGACY for uplink and downlink communication.** Our rationale was the following --- LEGACY can be used to send gradients from the client to the server and/or from the server to the clients. This is a widely adopted practice in the gradient compression community, in practice, all compressors can be used in uplink and downlink channels; e.g., see [“*Bidirectional compression for SGD*” in pp. 4 of **Natural Compression for Distributed Deep Learning, Horvath et al., Proceedings of Machine Learning Research vol 145:1–40, 2022**]. Therefore, we are unsure about the principle the reviewer referred to that might prevent using LEGACY for downstream compression. Could the reviewer please clarify their question better?
>
> **LEGACY with threshold sparsifier.** LEGACY can be combined with any compression technique. For integrating hard thresholding, LEGACY recommends selecting adaptive thresholds based on either training dynamics or layer sizes. LEGACY suggests that the threshold should be decreasing over the iterations so that later training phases get more severe compression.

---

> > ### Comment · Reviewer_KVdq · 2024-11-25
> >
> > Thank you for your response, but my concerns remain, so I will keep my original score.

---

> > > ### Author Response · Authors · 2024-11-25
> > >
> > > Can you please **objectively** mention what are your concerns? Which of them remain? We will try to incorporate them in our best capacity.
> > >
> > > Thank you.

---

### Official Review · Reviewer_qMHq · 2024-11-05

**Soundness:** 2
**Presentation:** 3
**Contribution:** 2
**Rating:** 3
**Confidence:** 4

**Summary:**

The paper introduces LEGACY, an adaptive framework for gradient compression in distributed deep neural network training. This framework dynamically adjusts compression parameters based on layer size or training phase, depending on the user choice, to leverage the observation that smaller layers and the initial stages of training are more sensitive to aggressive compression. The authors provide theoretical intuitions to justify the adaptive scheduling approach, showing that it aligns with the convergence behavior of compressed stochastic gradient descent. The authors benchmarked the proposed algorithm across six DNN architectures and five datasets with different parameters against Top-k and random-k sparsification.  In summary, the paper presents a novel and practical approach to adaptive gradient sparsification, offering a scalable and efficient solution for distributed deep learning training across various applications.

**Strengths:**

The paper touches upon a very important challenge in training neural networks. To make training more efficient and scalable, there has been tremendous work on gradient compression to tackle the communication bottleneck in the distributed setup. However, over the past few years, some solutions have been proposed beyond the one-fits-all strategy, and ways have been developed to identify the compression adaptively during the training and for each layer. Further exploring this direction would allow us to exploit gradient compression's power to accelerate fully.

The paper looks into two different properties of neural network training, the importance of layer size and training stages in gradient sparsification, aiming to make gradient compression even more efficient.

One of the main strengths of the proposed method is its lightweight design, allowing for easy integration with NCCL, a popular library for distributed training. This accessibility means that the method could be widely adopted in practice without adding significant complexity, a crucial advantage in large-scale applications.

Moreover, the authors offer valuable theoretical insights, explaining why higher compression might be especially effective in the later stages of training. This theoretical perspective strengthens the overall argument and adds a layer of rigor to the paper’s contributions, making it both informative and practical.

**Weaknesses:**

- The paper lacks benchmarking against a closely related work, the L-GreCo method by Markov et al. (2024), which also aims to find optimal parameters per layer at each iteration. For instance, Markov et al. reported that L-GreCo achieves 76.85% accuracy on ResNet50 on ImageNet with a 45.6x compression ratio. Including a comparison with this approach would provide a stronger baseline and help clarify the performance gains offered by LEGACY.

- Although the paper suggests that the method can integrate with other compression techniques like quantization and low-rank decomposition, all experiments focus solely on sparsification. Additional experiments with various compression methods would demonstrate the flexibility and versatility of LEGACY in handling different compression techniques.

- The experiments focus primarily on a single compression regime. Testing LEGACY across different compression levels—both high and low—would help evaluate its performance against uniform sparsification and adaptive compression methods in varied regimes.

- The motivation behind the framework relies on two observations: increasing compression towards the end of training and applying higher compression to larger layers. However, the framework allows only one of these settings to be chosen at a time. A combined approach that integrates both strategies could offer a more comprehensive solution.

- A potential way the method can be improved is by considering the sequential order in which gradients are sent during backpropagation. Since gradients are transferred layer by layer from the last to the first, adjusting compression based on this order, in addition to layer size and training phase, might further enhance efficiency. For instance, prioritizing higher compression on gradients for earlier layers could optimize transfer times, as these gradients are sent later in the process.


Updates after the rebuttals:

Legacy introduces a framework that adaptively selects compression ratios during training and across layers by specifying compression parameters for different training phases or layer groups. However, two closely related works leverage similar ideas.

The first is Accordion by Agarwal et al., which adjusts compression levels during training based on critical training regimes. Accordion uses gradient norms as a proxy for the importance of the training phase, practically leading to less compression at the start of training or after learning rate reductions—aligning closely with Legacy's first mode. When comparing the two, Legacy offers more granular training phases and compression levels (which could be integrated into Accordion via multiple gradient thresholds). Legacy is computation-free, whereas Accordion accumulates gradients, a process with minimal overhead. However, Legacy requires manual specification of training phases, offering flexibility at the expense of additional hyperparameter tuning.

The second is L-GreCo by Markov et al., which employs low-overhead dynamic programming to optimize layer-wise parameters during training. This method leverages varying sizes and layers' sensitivities. In comparison, Legacy enables grouping layers by size and assigning compression parameters per group, while L-GreCo requires a range of possible compression choices and a uniform baseline for error bounds to find each layer's compression level optimally. L-GreCo also accumulates gradients and uses an efficient dynamic programming step, which, as reported in Table 1 of their paper, typically adds less than 0.5% to training time.

Moreover, as shown in Figure 6.a. of L-GreCo, it is possible to combine the approaches of L-GreCo and Accordion for a training-phase- and layer-aware adaptive compression method that outperforms previous adaptive techniques. Legacy could also combine these observations with additional parameter choices.

While the authors emphasize that Legacy is computation-free and preferable in low-bandwidth federated settings, to my understanding, prior methods have negligible computational overhead, making this advantage less compelling.

Regarding scalability, I acknowledge the computational demands of large-scale experiments and wouldn't penalize the paper for lack of such experiments. However, I question whether trends observed on ResNet18 with CIFAR-100 are generalizable to more complex tasks and datasets.

In conclusion, Legacy builds on two previously explored observations and introduces additional hyperparameter requirements. Given this, I find the paper's contribution unclear and will maintain my score (not due to a lack of large-scale experiments). I welcome further discussion if needed and encourage the authors to refrain from personal remarks towards reviewers.

**Questions:**

- How should the parameters for defining the "begin" and "end" training stages, or "small" and "large" layers, be chosen, potentially to match a specific uniform sparsification baseline? How does the proposed method decide between different compression parameter sets?

- Have the authors considered incorporating factors beyond layer size, such as layer type, into their compression framework? Certain layers may have varying sensitivity to compression; for instance, in CNNs, initial convolutional layers are likely less sensitive than later layers that capture finer features. Would accounting for these sensitivity differences enhance the framework's adaptability and performance?

- There is limited guidance on how to select parameters for defining the "begin" and "end" training stages or for distinguishing between "small" and "large" layers, as both of these could vary for different models and training purposes. How the following parameters would be chosen for a new task?

- How does the model distinguish between different sets of parameters and decide which set is better? Should these parameters be treated as hyperparameters that require tuning?

- In Figure 1, smaller layers are described as having less compression, yet the figure indicates a higher compression ratio for these layers. Is there a specific reason for this or a possible mismatch between the figure and the intended interpretation?

---

> ### Author Response · Authors · 2024-11-20
>
> We thank the reviewer for asking some interesting questions. Below we answer them.
>
> **Benchmarking against L-Greco.** We respectfully disagree with the reviewer. We compared LEGACY with 4 state-of-the-art adaptive compression strategies, [CAT by Khiriratetal. (AAAI 2021) with 16 citations, Variance-based compression by Tsuzuku et al. (ICLR 2018) with 86 citations, Accordion by Agarwal et al. (MLSys 2021) with 24 citations, and AdaComp by Chen et al. (AAAI 2018) with 201 citations]. We were aware of L-GRECO, which was put in the ArXiv in October 2022 and cited 1 time since then. We believe we have enough baseline comparisons, and adding one more baseline is possible but is not needed to convey the message --- LEGACY outperforms all the approaches mentioned above. Nevertheless, after the reviewer mentioned, we compared LEGACY with L-GRECO on ResNet-18 training on the CIFAR-100 dataset and both adaptive strategies of LEGACY are at par or outperform L-GRECO by sending albeit similar data volume. Additionally, we note that in contrast to L-GRECO, LEGACY is compute-free --- it does not need to calculate optimal parameters per layer at each iteration.
>
> **Extra Experiments.** We postulated in the paper that LEGACY can be used conjointly with any compression techniques in designing its compute-free, adaptive counterpart. The present set of experiments show the robustness and ease-of-use of LEGACY in conjunction with not only with sparsifiers but with any other compression techniques justifying our statement in Lines 106-108.
>
> **Setup.** We consider two popular compression techniques: random dithering, also, popularly known as quantized SGD (QSGD [2]), and low-rank compression technique, PowerSGD [3].
>
> **QSGD experiment details.** We train ResNet-9 on the CIFAR-10 dataset by using 2 workers for 30 epochs. QSGD comes with *user-defined* parameters, $s\ge 1$ stands for quantization level and $l\in N$ with $0\le l<s$. For uniform QSGD, we set s=32 throughout the training. For LEGACY, we use 3 groups to represent three different training phases, the beginning of training (1-10 epoch, with s=64), the middle of training (11-20 epoch, with s=32), and the end of training (21-30 epoch, with s=16); and 4 groups to represent four distinct layer sizes, very small layer, S (<600),  that are left uncompressed, medium-sized layers, M (<100,000 with s=256), large layers, L (<1,000,000, with s=64) and huge layers, H $(\ge 1,000,000,$ with s=16).
>
> || Unifrom QSGD (s=32) |  LEGACY (epoch) | LEGACY (Layer) |
> |:---:|:-------------------------:|:---------------------:|:-------------------:|
> |Accuracy [Avg. Data volume]|  87.21% [18.76%]       | 87.97% [18.76%]  |88.42% [16.4%]   |
>
> **PowerSGD experiment details.** We train ResNet-18 on the CIFAR-100 dataset using 2 workers for 200 epochs. PowerSGD comes with one user-defined parameter, rank (r). The smaller the rank the more aggressive the compression is. For uniform PowerSGD, we set r= 3 throughout the training. For LEGACY, we use 4 groups to represent four different training phases, the first quartile of training Q1 (1-50 epoch, with r=6), the second quartile of training Q2 (51-100 epoch, with r=4), the third quartile of training Q3 (101-150 epoch, with r=3),  and the final quartile of training Q4 (150-200 epoch, with r=2); and 4 groups to represent four distinct layer sizes, very small layer, S (<600),  that are left uncompressed, medium-sized layers, M (<100,000 with r=8), large layers, L (<1,000,000, with r=3) and huge layers, H $(\ge 1,000,000,$ with r=2).
>
> |      | Uniform PowerSGD(rank=3)|   L-Greco          |  Accordion           | LEGACY (epoch)  |  LEGACY (layer)  |
> |:---:|:-------------------------:|:---------------------:|:-------------------:|:-------------------:|:-------------------:|
> |Accuracy [Avg. data volume] |        74.58 % [1.06%]    |   75.23% [0.93%]  |    75.11%[0.75%] |   75.55%[1.1%] | 75.21% [1.09%]|

---

> ### Author Response · Authors · 2024-11-20
>
> **Defining Parameters.** This is a great question. The target of the parameter tuning for LEGACY is to have the same data volume over the training as the base uniform compressor and the process is fairly simple. Let $\delta_1$ be the compression ratio for the beginning of the training and $\delta_2$ for the end, with $\delta_1>\delta_2$. Further, let $\delta$ be the uniform compression ratio. To maintain the equal data volume, we need to have $\delta = (\delta_1 + \delta_2)/2$. E.g., If the uniform compression ratio is $\delta=0.1$, then based on the above relation one can set, $\delta_1=0.15$ in the first half of the training and $\delta_2=0.05$ in the second half. Please see our numerical experiments Lines 419-427. In a similar spirit, if $\delta_s$ and $\delta_l$ correspond to the compression of small and large layers, respectively, then we can set $\delta = (S\delta_s + L\delta_l)/(S+L)$, where $S$ and $L$ denote the dimension of small and large layers, respectively. For a given $\delta$ and corresponding network specific, $S$ and $L$, we can reverse design $\delta_s$ and $\delta_l$. We will explain this in the main paper.
>
> **Question.** How does the proposed method decide between different compression parameter sets?
>
> **Answer.** We note that *there is no recipe for choosing compression parameters*; see our discussion in Lines 45-74. The choice of compression parameters depends on multiple factors such as the dataset used, DNN model architecture, network topology, network bandwidth, and many more; see [1] and references therein. In contrast to compute-heavy state-of-the-art adaptive compressors, LEGACY is based on two simple propositions: (a) the layer size of the DNNs influences in choosing how much one needs to compress, smaller layers have insignificant effect compared to large layers, and (b) the training phase of the DNNs can be a critical contributor in the adaptive compressor design, the end training phase can tolerate severe compression without any accuracy lost. While the layer sizes can be determined and grouped based on their relative sizes, the only rule for choosing compression parameters based on the training phase is to *choose decreasing compression parameters over iterations*; see **Extra Experiments** which validates this. In the main paper, for simplicity, we split the training into two phases; see LEGACY setup in Lines 411-417. However, one can be more innovative and have multiple training phases, with decreasing compression ratios, in each phase as done in **Extra Experiments.** We will add these results in the main paper.
>
> To conclude, *LEGACY does not involve any parameter tuning*. Its adaptive compression strategy is based on two fundamental factors in DNN training, layer size, and training phase, and the new experiments along with the experiments in the main paper establish its robustness.
>
> **Incorporating other interesting factors in LEGACY such as layer types.** This is certainly an interesting factor that could influence the performance. However, we currently lack theoretical evidence to support it. In this paper, we focused on factors such as training epochs, and layer size, for which we have theoretical insights and numerical evidence. Exploring this aspect is left for future work.
>
> **Guidance of Parameter Selection.** In the main paper, for simplicity, we split the training into two phases and layers into two groups, small and large; see LEGACY setup in Lines 411-417. However, a more fine-grained breakdown of each factor is possible; see our **Extra Experimental Results.** One can change the compression parameters at each iteration or each epoch or after some number of iterations. The same granularity holds for layer sizes. We recall the only rule for choosing compression parameters based on the training phase is to *choose decreasing compression parameters over iterations*. A similar rule applies to the layer size.
>
> **Y-axis label in Figure 1.** If compression ratio is $\delta$, then we denote $100\times (1-\delta)$ as the compression factor, as we are working with $\delta$-compressors, where $\delta\in(0,1]$; see definition in Lines 191-194. Therefore, the y-axis in Figure 1 should read as compression factor = (1-the compression ratio)*100. We will clarify this in the revised version.
>
> **Reference**
>
>  1. Hang Xu, Chen-Yu Ho, Ahmed M. Abdelmoniem, Aritra Dutta, El Houcine Bergou, Konstantinos Karatsenidis, Marco Canini, and Panos Kalnis. Grace: A compressed communication framework for distributed machine learning. In IEEE ICDCS, 2021.
>
> 2. Dan Alistarh, Demjan Grubic, Jerry Li, Ryota Tomioka, and Milan Vojnovic. QSGD: Communication efficient SGD via gradient quantization and encoding. NeurIPS, 2017.
>
> 3. Thijs Vogels, Sai Praneeth Reddy Karimireddy, and Martin Jaggi. PowerSGD: Practical low-rank gradient compression for distributed optimization. NeurIPS,2019.

---

> > ### Comment · Reviewer_qMHq · 2024-11-26
> >
> > Thank you for your detailed response and the additional experiments provided.
> >
> > LEGACY focuses on utilizing differences in layer sizes and training stages to compress gradients during neural network training. However, the paper does not propose an algorithm to systematically decide between or combine these two approaches. Additionally, there is no clear guidance on selecting compression parameters for early vs. late stages or for small vs. large layers. From my perspective, the main contribution of this paper is to demonstrate that considering differences in the importance of training stages or layer sizes can allow for more effective compression without significant performance degradation compared to a uniform baseline. However, these ideas have already been explored—Accordion has addressed stage-wise importance, L-GreCo has handled layer-size awareness, and existing works have also explored combining these techniques into a unified approach that accounts for both aspects.
> >
> > Furthermore, the paper does not include experiments evaluating how the proposed method scales with increasing GPUs or nodes. It is also unclear whether the trends observed on the CIFAR-100 dataset using ResNet-18 would generalize to modern deep learning architectures and more challenging tasks.

---

> ### Author Response · Authors · 2024-11-28
>
> We encourage the reviewer to read our paper, especially pp. 6 Section 3.2 and 3.3 meticulously before making general comments. We understand that it is not personal, but apparently, the reviewer is trying to find a way to hold things against our work with improper, superficial, and partial justification.
>
> LEGACY is executed through the intermediary API call chooseparam in Algorithm1 (see pp. 4) which is standard compressed distributed training without error feedback. Functions 2 and 3 in pp. 5 determine the changing compression level of the function ``chooseparam" in Algorithm 1 based on epoch-based or layer-based compression, respectively. Next, in Lines 317-323, we mentioned how these approaches can be combined; also, see the system architecture in Figure 2. In LEGACY, the user has the freedom to choose the parameters based on any training phase or layer size. While the layer sizes can be determined and grouped based on their relative sizes, the only rule for choosing compression parameters based on the training phase is to *choose decreasing compression parameters over iterations.* These are in sharp contrast with Accordion by Agarwal et al. where only two training regimes are defined. As our new experiments indicate, LEGACY can clearly incorporate multiple training regimes efficiently. Moreover, Accordion does not come with any theoretical insights, LEGACY provides a more general view with a theoretical guarantee, and the adaptive compression in LEGACY is computation-free; see our discussion in Section 5.4.
>
> We acknowledge that L-GreCO indeed takes layer size into consideration but there is no clear link between the compression degree and layer size as we have. L-GRECO remarks about the training regime, where the authors claimed that in their experiments, Accordion, in most cases, found the critical regime in the first few epochs, but there are no direct claims about the importance of layers based on their sizes.
>
> Finally, we mentioned in Lines 503-505 that “We also found that at the core, these methods exhibit similar behavior to our strategies, confirming the effectiveness of our approach, which does not require additional computation.” No other paper in adaptive gradient compression supported this insight with theory as LEGACY did in Section 3.1.
>
> **Multi-GPU experiments.** In our small-scale budget and university support, we only had access to 2-4 GPUs, but we already tested the scalability of LEGACY under more constrained environments. We trained ResNet-18 on CIFAR10 using 50 workers, sharing a 1Gbps network bandwidth, with every worker operating on an Intel Xeon Platinum 8276 CPU instead of a GPU; see this is for distributed setup in Section B.3 in Appendix, and federated setup in the main paper, in Section 5.5. Moreover, in Table 3, we reported the training configuration and number of independent runs, which show the performance of LEGACY is not a stochastic phenomenon.  The reviewer should comply and note that the ICLR reviewer guideline says, **“You can ask for additional experiments. New experiments should not significantly change the content of the submission. Rather, they should be limited in scope and serve to more thoroughly validate existing results from the submission.”** Therefore, we were happy to mitigate the reviewer’s confusion by considering **two popular compression techniques: random dithering or QSGD, and PowerSGD with LEGACY and compared LEGACY (by adding multiple training phases and increasing more varieties in layer size) with L-GRECO and Accordion.** In contrast, what the reviewer is asking now is unjustified. Authors cannot arrange overnight access to multiple GPUs. Therefore, it is not justified to hold not running multi-GPU scalability experiments against us; we validated the efficacy of LEGACY in a much-constrained setup.
>
> Finally, the reviewer commented that **”Existing works have also explored combining these techniques into a unified approach that accounts for both aspects.”** For this, we are not aware of any works that give direct remarks on the effect and the combination of those factors, theoretically and empirically. We sincerely request the reviewer to provide references to these works and educate us. Otherwise, as we mentioned, subjective comments like this are not justified grounds to reject a paper.

---

### Author Response · Authors · 2024-11-23
**Requesting the Reviewers in Engaging Discussion With the Authors ---- Rebuttals were posted on 20/November 2024.**

Dear Reviewers,

We posted our detailed rebuttals addressing your questions on the 20th of November 2024 around 2:30 PM EST. Indeed, the comments strengthen the paper. **In particular, we performed several extra experiments and comparisons as Reviewer qMHq mentioned. These experiments show the compatibility of LEGACY with any compressors, including quantization and low-rank methods, and not only sparsifiers as was the complaint before. Furthermore, we played with training phases and layer sizes.** We reestablished that the only rule for choosing compression parameters based on the training phase is to decrease compression parameters over iterations; a similar setting can be applied to layer sizes. To conclude, LEGACY does not involve any parameter tuning --- Its adaptive compression strategy is based on two fundamental factors in DNN training, layer size, and training phase, and the new experiments along with the experiments in the main paper establish its robustness.

We hope the reviewers have read our rebuttals and will engage with us in meaningful interaction. We will be happy to clarify any other doubts.

Thank you again for your time.

Looking forward,

Authors of LEGACY

---

### Note · Authors · 2025-01-22

I have read and agree with the venue's withdrawal policy on behalf of myself and my co-authors.